# Activation of M1 cholinergic receptors in mouse somatosensory cortex enhances information processing and detection behaviour

Wricha Mishra [1], Ehsan Kheradpezhouh[1,2] & Ehsan Arabzadeh [1,2]✉

To optimise sensory representations based on environmental demands, the activity of cortical neurons is regulated by neuromodulators such as Acetylcholine (ACh). ACh is implicated in cognitive functions including attention, arousal and sleep cycles. However, it is not clear how specific ACh receptors shape the activity of cortical neurons in response to sensory stimuli. Here, we investigate the role of a densely expressed muscarinic ACh receptor M1 in information processing in the mouse primary somatosensory cortex and its influence on the animal's sensitivity to detect vibrotactile stimuli. We show that M1 activation results in faster and more reliable neuronal responses, manifested by a significant reduction in response latencies and the trial-to-trial variability. At the population level, M1 activation reduces the network synchrony, and thus enhances the capacity of cortical neurons in conveying sensory information. Consistent with the neuronal findings, we show that M1 activation significantly improves performances in a vibriotactile detection task.

[1] Eccles Institute of Neuroscience, John Curtin School of Medical Research, The Australian National University, Canberra, Australia. [2]These authors jointly supervised this work: Ehsan Kheradpezhouh, Ehsan Arabzadeh. ✉email: ehsan.arabzadeh@anu.edu.au

To survive, animals need to process the arriving sensory information differently depending on the context; while the sound of rustling leaves may not be important in the burrow, it could warn a mouse of an approaching predator in the open field. Neuromodulators such as Acetylcholine (ACh) provide one mechanism through which animals fine-tune sensory processing to reflect the demands of the environment[1,2]. The cholinergic system modulates information processing across different cortical areas influencing the animal's behavioural state and level of attention[1,3,4].

The cholinergic system is well suited to coordinate neural activity across different modalities, as it provides a widespread and diffuse innervation of the cortex[5]. Acting through various subtypes of ACh receptors (AChRs), ACh controls neuronal excitability and firing rate, by hyperpolarising or depolarising target neurons[6,7]. Cholinergic stimulation also reduces noise correlations and membrane potential fluctuations[8,9]. These functions are predominantly mediated by postsynaptic muscarinic AChRs (mAChRs) that activate the $G_{\alpha q}$-signalling cascade and consequently increasing neuronal excitability[10].

Consistent with these functions, activation of mAChRs is known to enhance the sensory representations across modalities of vision, audition and somatosensation[4,11–13]. As a well-studied example, ACh increased the response of neurons in the visual cortex to stimulus contrast[13] and enhanced orientation tuning of cortical neurons[14]. In the somatosensory cortex, mAChR modulation depolarised neuronal membrane potentials, and enhanced stimulus-evoked responses to deflections applied to the whiskers[11].

Despite growing evidence on the muscarinic neuromodulation, it is not clear how mAChRs shape the encoding of sensory inputs in single neurons and neuronal ensembles, ultimately determining the perceptual responses to those inputs. Here, we combine pharmacological manipulations with in vivo electrophysiology, 2-Photon Calcium ($Ca^{2+}$) imaging and behavioural studies, to characterise how activation of M1 receptors affects sensory information processing and perception.

We employed the mouse primary vibrissal somatosensory cortex (vS1) as it provides an optimal model to investigate neuronal coding due to its functional efficiency[15], structural organisation[16] and ecological relevance[17,18]. We demonstrate that M1 activation enhances the sensory evoked responses in the mouse vS1 neurons through a multiplicative gain modulation. We also show an M1-induced reduction in the first spike latency and the trial-to-trial variability in the evoked responses. We show that activating M1 induces desynchronisation in a subpopulation of neurons, reminiscent of attentive states[19]. Finally, we show that consistent with our neuronal findings, M1 activation significantly improves the ability of the mice in detecting vibrotactile stimuli applied to their whiskers. Together, these results depict a key role for M1 receptors in modulating sensory processing and behavioural sensitivities.

## Results

**M1 activation enhances evoked responses in vS1 neurons.** We first characterised the expression of M1 across layers of vS1 through immunostaining (Supplementary Figs. 1, 2). Consistent with previous observations[20], we found that M1 is prominently expressed in layers 2/3 and 5. We further identified that M1 is highly co-localised in the excitatory neurons (Supplementary Fig. 1; colocalisation with CaMKII; Pearson correlation coefficient in layer 2/3 = 0.93, Pearson correlation co-efficient in layer 5 = 0.89). To determine the effect of M1 modulation on sensory processing, we first performed loose cell-attached recording (juxtacellular configuration[21]) under urethane anaesthesia. We

recorded the activity of vS1 neurons in layer 2/3 and 5 during local activation or inhibition of M1 using a paired pipette method (Fig. 1a). We recorded and labelled vS1 neurons under continuous application of artificial cerebrospinal fluid (aCSF, control), M1 potentiator (Benzyl Quinolone Carboxylic acid, BQCA, 10 μM) or M1 specific inhibitor (Telenzepine Dihydrochloride, TD, 1 μM), while we stimulated the contralateral whiskers. The stimuli consisted of a brief vibration (20 ms in duration), which was presented at 5 different amplitudes (0–200 μm). We investigated the sensory evoked responses in the contralateral vS1 under these three conditions.

Figure 1b illustrates the effect of M1 modulation on the spiking activity in response to a 200 μm whisker deflection, recorded from an example neuron. M1 activation (BQCA) profoundly enhanced the stimulus-evoked responses, with no evident changes in the baseline activity (Fig. 1b, green). Subsequent application of M1 inhibitor (TD) reduced the evoked response of this neuron back to its initial level (Fig. 1b, magenta). Figure 1c illustrates how M1 modulated the response of the example neuron for the full range of stimulus amplitudes. Here, M1 activation produced an upward shift and M1 inhibition produced a downward shift in the response profile of the example neuron.

We observed a similar effect of M1 modulation across all recorded neurons (Fig. 1d, $n = 18$ in Layer 5, $n = 5$ in Layer 2/3, 17 mice, $p < 0.001$, Friedman test with Dunn's multiple comparison). Figure 1e demonstrates the effect of M1 modulation on three main parameters of the neuronal response function: the baseline activity (amplitude = 0 μm), the maximum evoked response, and the response range (the difference between the maximum and minimum responses) as a measure of coding capacity. BQCA did not modulate the baseline activity ($p = 0.17$, Friedman test with Dunn's multiple comparison; Fig. 1e, left panel), but significantly increased the maximum evoked response ($n = 23$, $p < 0.0001$, Friedman test with Dunn's multiple comparison, Fig. 1e, middle panel). Subsequent introduction of TD decreased the maximum evoked response to the initial values ($n = 23$, $p < 0.001$, Friedman test with Dunn's multiple comparison, Fig. 1e, middle panel). We observed a similar trend in the response range; BQCA significantly increased the response range ($n = 23$, $p < 0.001$, Friedman test with Dunn's multiple comparison) and TD reduced it back to its initial values (Fig. 1e, right panel). These results indicate that M1 activation enhances the representation of vibrotactile inputs in vS1 neurons. To further validate the findings, we performed neuronal recordings using a different M1 receptor agonist (Cevimeline Hydrochloride, 5 μM) and antagonist (Dicyclomine Hydrochloride, 5 μM). The agonist and antagonist produced a qualitatively similar modulation of neuronal activity as observed earlier with BQCA and TD. As before activation of M1 increased baseline firing rate, increased the maximum evoked response and enhanced the response range (Supplementary Fig. 3). In the following section, we investigate the effect of M1 modulation on other parameters of neuronal response including the latency and trial-to-trial variability.

**Temporal sharpening in vS1 neurons with M1 activation.** The reliability of neuronal response and its timing can reflect the behavioural relevance of the stimulus; faster (reduced latency) and more reliable responses (less variability across presentations) suggest enhanced detectability at the neuronal level and ultimately better coding efficiency[22,23]. On the other hand, higher variability in responses is detrimental to coding efficiency[24]. The intrinsic variability in the response can be modulated by sensory stimuli[25] and non-sensory parameters including neuromodulation[24,26]. Here, at the highest stimulus amplitude (200 μm deflection), we quantified the latency of the first evoked response. M1 activation

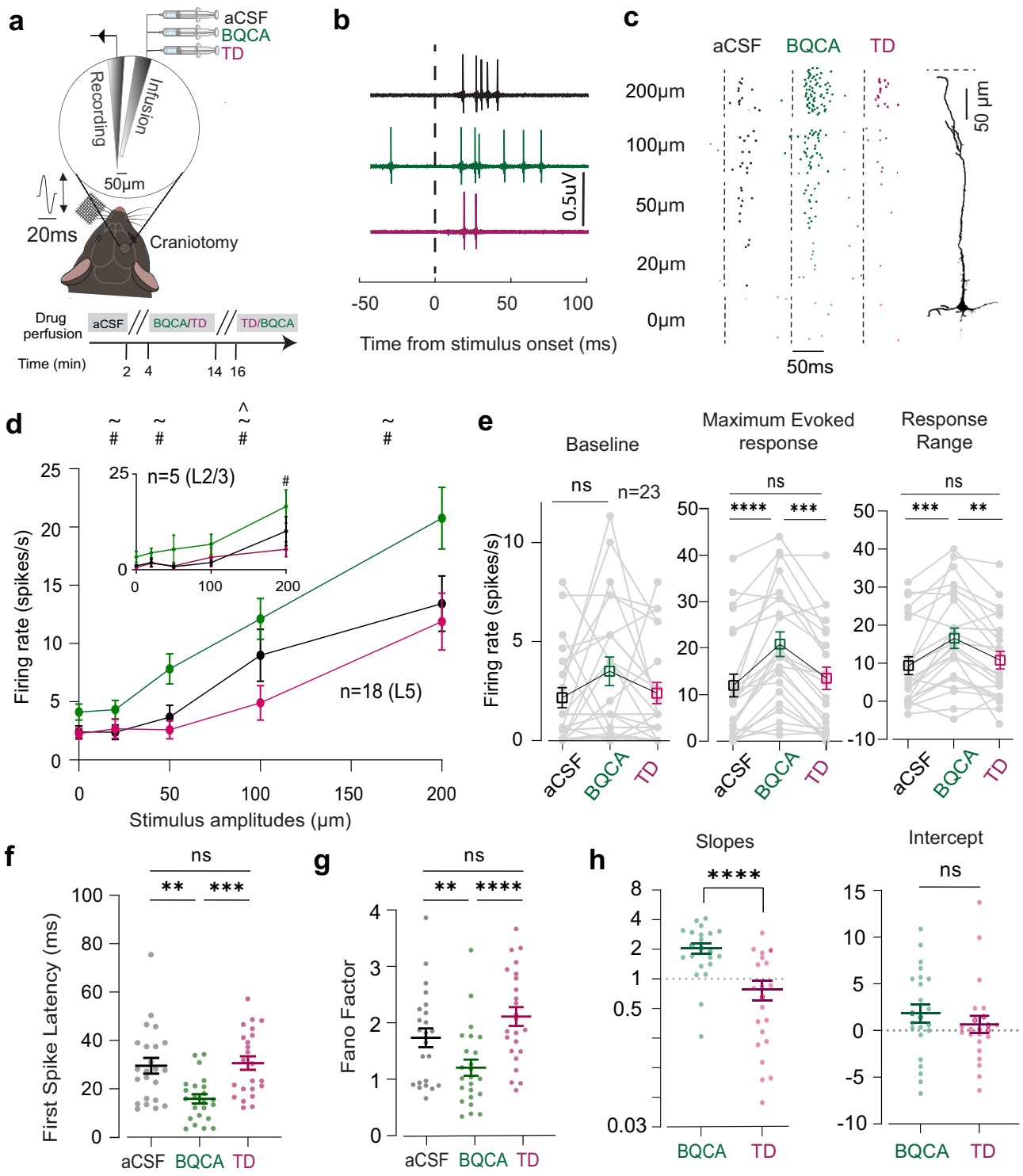

with BQCA significantly reduced the first-spike latencies compared to the control condition (Fig. 1f, $n = 23$, 17 mice, $p = 0.0035$, Friedman test with Dunn's multiple comparison); along with reduced jitter (Supplementary Fig. 4a, as seen by reduced Standard Deviation, $p < 0.001$, Wilcoxon signed-rank test). The latencies increased by subsequent application of TD (Fig. 1f, $p = 0.0004$, Friedman test with Dunn's multiple comparison).

To quantify the reliability of the evoked responses, we used Fano factor (the ratio of the variance to the mean of the firing rate) as a measure of trial-to-trial variability. A Fano Factor close to 1 reflects a Poisson distribution where the mean and variance of the response are the same[27]. A higher Fano factor indicates a less reliable response from one trial to another (lower coding efficiency, Adibi et al.[28]). We applied this analysis to the evoked response at all stimulus amplitudes (Supplementary Fig. 4b). At the highest stimulus amplitude, M1 activation significantly decreased the mean Fano factor (Fig. 1g, $p = 0.0045$, Friedman test with Dunn's multiple comparison; aCSF versus BQCA);

**Fig. 1 M1 activation enhances evoked responses in vS1 neurons. a** A schematic of the juxtacellular electrophysiology set-up for pharmacological manipulation of M1. The magnified circle depicts the custom-made pipette pair and the bottom panel depicts the drug perfusion protocol. **b** Raw voltage traces showing the spiking activity for an example neuron in response to a 200 μm whisker vibration after application of aCSF (black, control), BQCA (green, M1 agonist) and TD (magenta, M1 antagonist). The black vertical dotted line represents the stimulus onset. **c** Raster plots of spiking activity for an example neuron after application of aCSF, BQCA and TD. The y-axis shows the trial numbers at subsequent presentations of the same stimulus. Inset: The reconstructed image shows the morphology of the example neuron, which is a layer 5 thick-tufted pyramidal neuron. The scale bar is 50 μm; the horizontal dotted line represents the dura. **d** The input/output function for all neurons under aCSF, BQCA and TD conditions. Each dot represents the mean firing rate across neurons ($n = 18$ in Layer 5, $n = 5$ in Layer 2/3, --Significant statistical difference between BQCA and control; #- between BQCA and TD; ^- between TD and control. **e** M1-induced changes in baseline firing, evoked response and the response range. Grey dots indicate single neurons and the squares represent the means ($n = 23$, 17 mice). **f** First spike latencies calculated in a 100-ms window post stimulus presentation (200 μm, $n = 23$). **g** Fano factors of the evoked firing rate (200 μm). **h** Left: The slopes of the best-fitted lines of neuronal response after M1 activation and inhibition. ($n = 23$, ****$p < 0.0001$, Wilcoxon signed-rank test). Right: The y-intercepts of the best-fitted lines. Each dot represents the slope or y-intercept of a neuron ($n = 23$, $p > 0.05$, Wilcoxon signed-rank test). For (**f–h**), the horizontal and vertical lines represent the mean and SEM respectively. For (**e–g**), **$p < 0.01$, ***$p < 0.001$, ****$p < 0.0001$, Friedman test with Dunn's multiple comparison.

subsequent M1 inhibition increased the average factor (Fig. 1g, $p < 0.0001$, Friedman test with Dunn's multiple comparison; BQCA versus TD). These results show that M1 activation reduced the trial-to-trial variability in the evoked spike count among neurons, which indicates an increase in reliability. Using these data, we next investigate how these M1 modulations further affect the encoding of stimulus features.

**Multiplicative gain modulation in neuronal response function through M1 AChR.** For every stimulus amplitude, we plotted the response under BQCA or TD condition against that response under aCSF condition; we calculated the slope and intercept of the best-fitted line for each neuron (Supplementary Fig. 4d). The slope and intercept provide information about the effect of M1 modulation on coding efficiency[29]. The slope illustrates multiplicative changes in the neuronal response function and the y-intercept illustrates the additive changes. For example, a slope of 1 with a positive y-intercept would indicate an additive function signifying a consistent increase in response across all stimulus amplitudes. Conversely, a slope higher than 1 would indicate a multiplicative gain modulation signifying that the increase in activity is multiplicatively scaled from lowest to highest stimulus amplitudes.

Our results were consistent with an M1-induced multiplicative gain modulation in the response function. Overall, 91% of the recorded neurons exhibited a slope greater than 1 with a mean of $2.05 \pm 0.23$ (Fig. 1h, green, $p < 0.0001$, Wilcoxon signed-rank test). On the other hand, M1 inhibition reduced the gain of the response function, with an average slope lower than 1 ($0.78 \pm 0.17$, Fig. 1h, magenta, $p < 0.0001$, Wilcoxon signed-rank test). The y-intercepts did not show a systematic change with M1 activation or inhibition ($p > 0.05$, Fig. 1h).

**Modulation of neuronal population activity by M1.** We next determined the effect of M1 modulation on the population activity of cortical neurons through 2-photon $Ca^{2+}$ imaging in both awake and anaesthetised mice (Fig. 2 and Supplementary Fig. 3). M1 is a $G_q$-protein coupled receptor that increases intracellular $Ca^{2+}$ through activation of various signalling cascades[30]. Here, we expressed GCaMP7f (a highly sensitive $Ca^{2+}$ sensor[31]; in layer 2/3 vS1 neurons. To modulate M1 activity, we implanted a cannula semi-parallel to the cranial window to perfuse the transfected area (see methods, Fig. 2a). Similar to previous experiments, we captured the effect of M1 modulation on the spontaneous activity and the evoked responses under aCSF (Control), BQCA (M1 activation) and TD (M1 inhibition) conditions. We then calculated changes in fluorescence ($\Delta F/F_0$) as a measure of neuronal activity.

Figure 2c captures $\Delta F/F_0$ in an example neuron (pointed with arrow, Fig. 2b) when the whiskers were stimulated at 250 μm amplitude in an awake head-fixed mouse. For this example neuron, M1 activation enhanced the $\Delta F/F_0$ and subsequent M1 inhibition reduced the $\Delta F/F_0$ to its initial values. These findings were consistent across the neuronal population (same mouse, Fig. 2d, $n = 60$).

We further quantified the effect of M1 modulation on neuronal population activity across all stimulus amplitudes. As observed in the example mouse, M1 activation significantly increased the sensory evoked response (Fig. 2e, $n = 944$, 6 mice, $p < 0.01$ for 50, 100 and 250 μm amplitudes); highest modulation was observed at 250 μm amplitude (Fig. 2f, right panel, $p < 0.001$). Consistent with the electrophysiological data (Fig. 1e, right panel), we did not observe any significant modulation in the baseline activity (Fig. 2f, left panel, $p = 0.07$, Friedman test with Dunn's multiple comparison). Interestingly, we found a subpopulation of neurons, which remained silent (no evoked response) under the control condition (aCSF), but became significantly responsive to the stimuli after M1 activation (Fig. 2d, Neuron # 30–55). Across all imaged neurons, 25% exhibited a significant sensory evoked response under the control condition (aCSF). This proportion increased to 31% under BQCA and returned to 23% after application of TD. Collectively, the increased number of responsive neurons and the magnitude of their evoked response support our previous findings that local activation of M1 enhances sensory representations in vS1.

**Enhanced synchrony in vS1 neurons with M1 inhibition.** We next investigated how M1 modulation affects the correlation of activity across recorded neurons. The capacity of a network to represent sensory information depends on the strength of the response to sensory stimuli (signals) as well as the similarity of responses in the absence of stimuli (noise correlation, Minces et al.[12]). This similarity between stimulus-independent responses (noise correlations) can limit the encoding capacity[22,32–34] and cholinergic modulation has been shown to affect noise correlations[12].

Here, we investigated how stimulus-independent noise correlations are affected by M1 modulations. Figure 3a shows the fluorescence traces from 6 example neurons (Neurons 1–6 numbered in Fig. 2b) along with their correlograms (Neuron 5: Neuron 6, bottom) under the aCSF, BQCA and TD conditions. We observed high levels of correlation between example neuron pairs (most evident between Neuron 5 and 6) after M1 inhibition (Fig. 3a, bottom right) and this correlation was reduced by activating M1 (Fig. 3a, bottom centre). This finding generalised to all recorded pairs; M1 inhibition with TD showed a significant increase in pairwise correlation (Fig. 3b, $p < 0.01$, 36 neuron pairs,

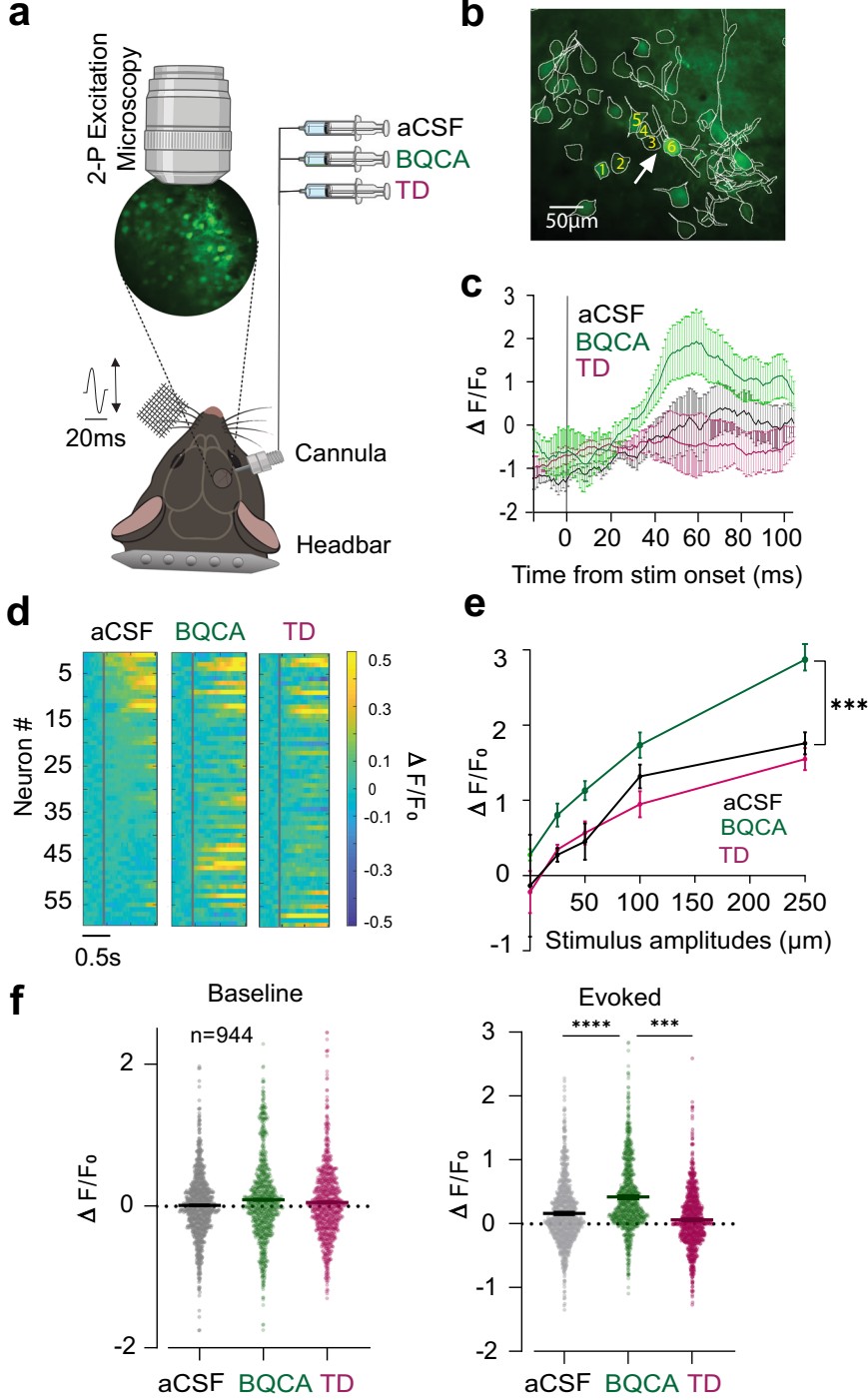

**Fig. 2 Characterisation of M1 modulation on neuronal population using 2-photon Ca$^{2+}$ imaging. a** A schematic depicting the Ca$^{2+}$ imaging setup for pharmacological manipulation of M1. The magnified circle depicts a 3 mm cranial window expressing GCaMP7f in vS1. The changes in fluorescence activity were recorded from the neuronal population while applying vibrations to the contralateral whisker pad. **b** A 2-photon image of layer 2/3 neurons expressed GCaMP7f after motion correction with all ROIs highlighted by a white outline. **c** Changes in fluorescence ($\Delta F/F_0$) in response to the 250 μm whisker deflection, under the control (aCSF), M1 activation (BQCA) and M1 inhibition (TD) conditions for the ROI shown in (**b**) (white arrow, $n = 30$ trials). The grey vertical line indicates the stimulus onset. Shaded bars indicate standard error of the mean $\Delta F/F_0$ across 30 trials. **d** A heatmap depicting the fluorescence activity of neurons shown in (**b**) ($n = 60$). The neurons in the aCSF (left panel) condition are sorted in a descending order and the sorting indices are conserved across BQCA (middle panel) and TD (right panel). Every horizontal line represents the $\Delta F/F_0$ of the same neuron across the three conditions, and the red line indicates the onset of a 250 μm whisker stimulation. **e** Changes in $\Delta F/F_0$ measured in a 1 s window after stimulus onset in the aCSF, BQCA and TD conditions for all stimulus amplitudes ($n = 944$, 6 mice). The dots are the mean $\Delta F/F_0$; the error bars are the SEM. **f** Baseline (left) and maximum evoked (right) neuronal response. Every dot represents a neuron. Black lines and error bars indicate the means and SEM ($n = 944$, 6 mice). ***$p < 0.01$; ****$p < 0.001$, Friedman test with Dunn's multiple comparison.

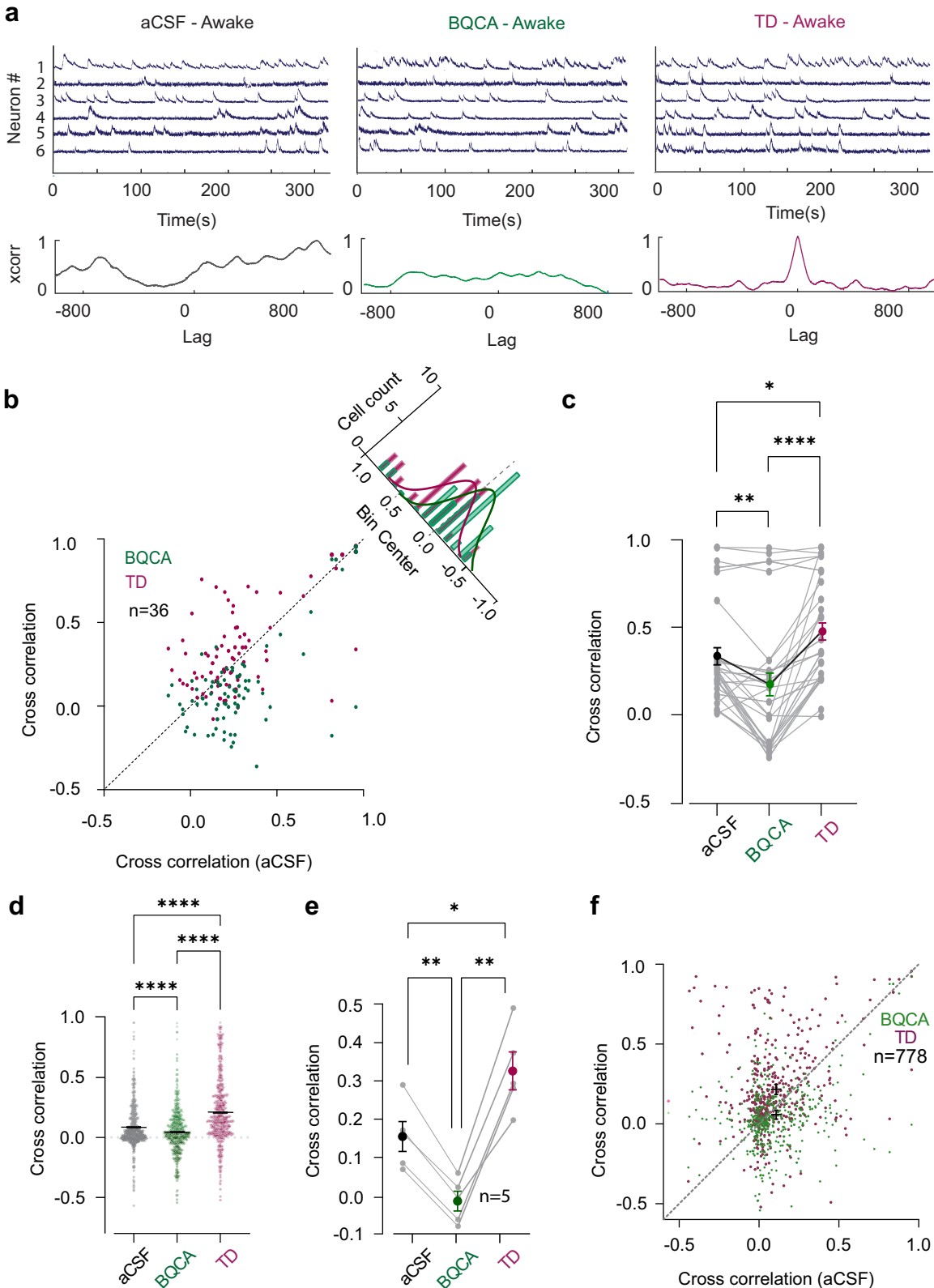

Wilcoxon signed-rank test); and M1 activation with BQCA reduced pairwise correlations in the majority of the pairs (Fig. 3b, green, $p < 0.0001$, 36 neuron pairs, Wilcoxon signed-rank test), resembling desynchronised activity. This trend could also be quantified across the neuron pairs shown in Fig. 3a (Fig. 3c, $p = 0.0057$, Friedman test with Dunn's multiple comparison). To

further visualise the relative contribution of signal and noise correlations, Supplementary Fig. 6 illustrates the changes in correlated activity in the pre-stimulus time period as well as in the presence of sensory stimuli using a joint-PSTH analysis[35].

We applied this analysis to all responsive neurons among all animals and sessions; by activating M1, an enhanced

**Fig. 3 The effect of M1 modulation on neuronal synchrony. a** Raw fluorescence traces of 6 example neurons from one session in Fig. 2b over time under control (aCSF), M1 activation (BQCA) and M1 inhibition (TD) conditions. The spiking activity is more synchronised by inhibiting M1 (TD). The correlograms (below insets) show the cross correlation values versus lag for an example neuron pair (neuron 5: neuron 6). The sharp peak in the cross correlation values after TD perfusion indicates greatest synchrony between the example neuron pair at lag zero. **b** Cross correlation of neuron pairs (for the 6 neurons shown in (**a**), under the aCSF condition versus BQCA (green) or TD (magenta) condition. Neurons after M1 inhibition show an increased noise correlation ($p < 0.01$) as compared to M1 activation ($p < 0.0001$, $n = 36$, Wilcoxon signed-rank test). Inset: Histogram distribution for correlations after M1 activation (green) and M1 inhibition (magenta) from the line of equivalence. **c** Average correlation coefficients in vS1 neurons in one mouse after aCSF, BQCA and TD perfusions ($n = 36$ pairs, $**p < 0.01$, $****p < 0.0001$, Friedman test with Dunn's multiple comparison). Mean correlation coefficients are indicated by dots of black, green and magenta. **d** Correlation coefficients across 5 mice with aCSF, BQCA and TD perfusions ($n = 778$, 5 mice, $****p < 0.0001$, Friedman test with Dunn's multiple comparison). Each dot represents the correlation coefficient of one neuron pair. The black bars indicate the mean correlation coefficients for each condition and the error bars represent SEM. **e** Mean cross correlations in the aCSF, BQCA and TD conditions across mice ($n = 5$, $*p < 0.05$, $**p < 0.01$, Friedman test with Dunn's multiple comparison). **f** Cross correlation across all animals and recording sessions, between aCSF, BQCA or TD. Neurons after M1 inhibition show increased noise correlation as compared to M1 activation ($n = 778$, 5 mice). The dashed grey line indicates the line of equivalence.

desynchronisation was systematically observed (Fig. 3d, $n = 778$, 5 mice, $p < 0.0001$, Friedman test with Dunn's multiple comparison). It is known that during desynchronised states, sensory information processing is enhanced both at the level of single neurons and in neuronal populations[12,36–39]. Overall, these results support our previous findings that M1 activation improved information transmission across vS1 neurons. The enhancement in the sensory evoked response in vS1 neurons and the desynchronisation following M1 activation led us to investigate the effect of M1 modulation on mouse detection performance.

**M1 modulation enhances detection performance**. Both electrophysiology and Ca$^{2+}$ imaging experiments demonstrated an M1-induced gain modulation of vS1 neurons. The enhanced sensory representations were evident at the level of single neurons (Fig. 1c, e, and Fig. 2d) and at the population level (Fig. 1d, Fig. 2e). The M1-induced gain modulation also resulted in an increased number of vS1 neurons that responded to the whisker input (25% under control to 31% under M1 activation). We therefore hypothesised that the enhanced representations in the vS1 would improve the mouse's ability to detect whisker stimuli. To determine the effect of M1 modulation on detection behaviour, we tested a simple detection task in awake head fixed mice while modulating M1 activity. As nocturnal animals, mice regularly use their whiskers to navigate and explore their surroundings. Depending on the behavioural state of the animal (active engagement or quiet wakefulness), the efficiency of sensory information processing is altered by neuromodulatory inputs like ACh[8]. Here, we investigated whether the observed enhancement in sensory processing through M1 receptors is reflected in the behavioural performance of mice.

Using a similar modulation method to that used in the electrophysiology and imaging experiments, we implanted a cannula in the right vS1 of 6 mice and attached a headbar to allow head fixation. After recovery, mice were trained to perform a whisker vibration detection task (Fig. 4a). Vibrations of different amplitudes were presented through a piezoelectric stimulator on the left whisker pad at amplitudes of 0, 15, 30, 60, or 120 μm. Mice received a sucrose reward for licking the spout on trials with vibration (15, 30, 60, and 120 μm) within a 1-s window; licking in the absence of vibration (0 μm) was not rewarded (Fig. 4b). Stimuli were presented as blocks of 5 trials, containing 4 vibration amplitudes (15, 30, 60, and 120 μm) and a no-vibration trial (0 μm) in a pseudorandom order. This allowed us to calculate detection rates within each block (Methods, Behavioural analysis). To allow the collection of a sufficient number of trials, only one solution was applied in a single behavioural session. As with earlier experiments, the solutions were aCSF (control),

BQCA (10 μM, M1 activation) or TD (5 μM, M1 inhibition) and were perfused through the implanted cannula. These sessions were pseudo-randomly intermixed and each session was repeated 5 times. This produced an average 240 trials per condition (48 blocks X 5 stimulus amplitudes). We found that the lick rate for 0 μm stimulus trials was similar to the pre-stimulus lick rate, indicating that mice successfully refrained from licking the spout in the absence of whisker vibrations (Fig. 4b; darkest line). As a general trend, mice licked at a higher rate and showed faster response times as the vibration amplitude increased (sample mouse, Fig. 4b, left inset, aCSF).

Consistent with our findings at the neuronal level, we observed that BQCA improved the post-stimulus lick rate (lick rates in sample mouse, Fig. 4b, middle inset, BQCA) whereas TD reduced the lick rate (sample mouse, Fig. 4b, right inset, TD). To better quantify the effect of M1 on detection, we compared the average response to the stimuli and the response time (the time of first lick after stimulus onset) among aCSF (control, black), BQCA (M1 activation, green) or TD (M1 inhibition, magenta) conditions (Fig. 4c). M1 activation significantly increased lick rates across all stimulus amplitudes (except 0), with the most significant rise observed at the highest amplitude (120 μm, green, Fig. 4c, $n = 6$; $p < 0.01$, Friedman test with Dunn's multiple comparison). The lick rate decreased under TD condition even below the control rate (magenta, Fig. 4c, $n = 6$; $p < 0.05$). In line with previous studies[32], we observed a reduction in the response time with increasing stimulus amplitude (Fig. 4d). The response (first lick) time for the highest stimulus amplitude decreased significantly after M1 activation (Fig. 4d, Inset, green; $p < 0.05$); M1 inhibition with TD increased this first lick time across all mice (Fig. 4d inset, magenta; $p < 0.01$, Friedman test with Dunn's multiple comparison).

To further investigate the effect of M1 modulation on perceptual sensitivity we quantified the detection rate for each mouse and condition (Methods, Behavioural analysis). As expected, the detection rates were generally higher at the beginning of a session (0–10 trials) and then gradually tapered towards the end of the session (Fig. 4e, black). As illustrated in the example mouse, activation of M1 produced a consistently high detection rate across all mice that was maintained for a higher number of trials (Fig. 4e, f, green) as compared to the control (Fig. 4e, f, black). The false alarm rates (response to no stimulus trials) were not significantly altered by M1 modulation (Supplementary Fig. 7b, $p > 0.05$, Friedman test with Dunn's multiple comparison), indicating that the enhanced detection rate is not a result of an overall increased lick rate. This increased sensitivity is directly captured in the d-prime measures (Fig. 4g). For every mouse, M1 activation (BQCA) enhanced the average performance (Fig. 4g; d-prime, aCSF: $2.08 \pm 0.21$; d-prime, BQCA:

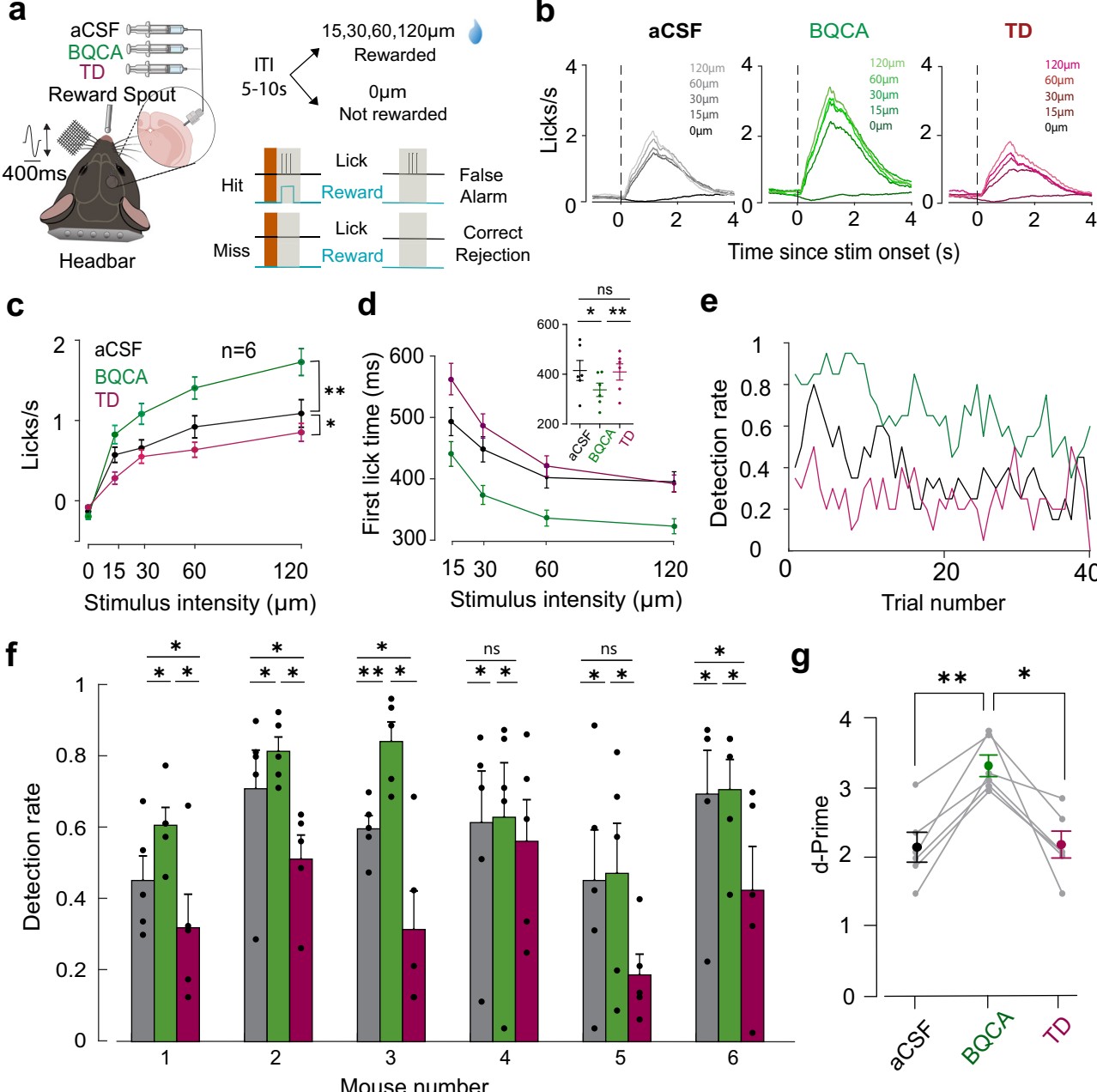

**Fig. 4 M1 modulation enhances vibriotactile detection behaviour. a** A schematic depicting the behavioural paradigm. Inset: A 400-ms, 40-Hz vibration stimulus was presented at amplitudes: 0, 15, 30, 60, or 120 μm. Licking the spout during the reward window of 1 s during stimulus presentation trials resulted in sucrose reward. Each stimulus presentation had an inter-trial interval of 5–10 s. **b** The licking profile of an example animal showing the average lick rate against time, with the stimulus onset marked by the vertical black dotted line. Different stimulus amplitudes are depicted in different shades. **c** The lick rates across all stimulus amplitudes (0, 15, 30, 60, or 120 μm) in the control (black), M1 activation (BQCA, green) and M1 inhibition (TD, magenta) conditions. The solid dots represent the mean lick rates and the error bars represent SEM. ($n = 6$, $*p < 0.05$, $**p < 0.01$, Friedman test with Dunn's multiple comparison). **d** Average first lick time (response time) across stimulus amplitudes for the control, BQCA and TD conditions. Inset: The first lick time across all 3 conditions for the highest stimulus amplitude of 120 μm ($*p < 0.05$, $**p < 0.01$, Friedman test with Dunn's multiple comparison). **e** Detection rate calculated across a session of 40 trials blocks (each block consists of 5 trials) for an example animal under the control, M1 activation (green) or M1 inhibition condition (magenta). **f** Average detection rate across 6 animals in (**c**) under the control, M1 activation and M1 inhibition conditions. M1 activation significantly enhanced detection rate and inhibiting M1 decreased this detection back to baseline levels across animals. Mean detection rate for every animal is averaged over 5 sessions in each condition ($*p < 0.05$, $**p < 0.01$, Friedman test with Dunn's multiple comparison). **g** Mean d-prime across 6 mice: 2.08 (aCSF), 3.26 (BQCA), 2.12 (TD, $**p < 0.01$, $*p < 0.05$, Friedman test with Dunn's multiple comparison). For (**f**) and (**g**) the vertical errorbars indicate the SEM.

3.26 ± 0.15) whereas, blocking M1 (TD) reduced the average performance to the control values (Fig. 4g, d-prime, TD: 2.125 ± 0.19). All together, these findings suggest that M1 activation improved perceptual sensitivity as was reflected in faster and more reliable responses.

## Discussion

To adjust the animal's behavioural state to the demands of the environment, cortical activity is regulated by neuromodulators including the cholinergic system. Cholinergic input to the cortex has long been considered to act as a global activating system[40,41]. In particular, layer 2/3 pyramidal neurons are powerfully influenced by ACh through the dense projections they receive from the basal forebrain[42]. Layer 2/3 is considered as a hub in cortical processing[16,43], where the majority of neurons fire sparsely due to a balanced feedforward excitation and feedback inhibition[44]. ACh is thought to modify this balance to alter cortical activity and to shape the flow of information within the cortical circuits. Here, we showed that M1 activation significantly enhanced the sensory-evoked responses and reduced the trial-to-trial variability of these responses. This was illustrated by an increase in the signal strength, a decrease in the Fano Factors of the firing rates and a decrease in the first-spike latencies. At the population level, M1 activation reduced the network synchrony, which in turn enhanced the capacity of vS1 neurons in conveying sensory information. Consistent with the neuronal findings, we found that M1 activation improved performance in the vibriotactile detection task. Together, these findings show that M1 receptors enhance information processing in the vS1 and this reflects in the animal's ability to better detect sensory inputs. Our method of activating M1 through a local potentiator can be considered as a phasic release of ACh mediated by BF neurons, at the scale of seconds. Our findings are thus consistent with phasic ACh release and a volume transmission hypothesis as the circuit mechanism underlying cognitive operations such as attention[45]. Attention is known to dynamically change sensory representations in the cortex, with increased attention leading to an improved signal-to-noise ratio. The attentional modulations of sensory representations determine how we discriminate between stimuli[46] and integrate multiple sensory inputs[47]. Previously, it has been shown that activating the cholinergic system enhances neuronal responses to sensory stimuli in a way that resembles a gain modulation[13,48]. Such gain modulations reflect the changes in the sensitivity of a neuron to the stimulus while its selectivity for that particular stimulus is preserved[49] For example, neurons receiving a wide range of stimuli must be sensitive to weak stimuli but not saturated in response to stronger ones. In this way, neuronal responses can be continuously modulated to process sensory inputs at a wide dynamic range[50]. Gain modulation causes an upward shift in the neuronal response function and strengthens the representation of sensory stimuli[4,51]. Here, we showed that M1 activation enhanced the evoked responses in vS1 through a multiplicative gain modulation. Consistent with previous literature, activation of M1 led to changes in neuronal sensitivity, which in turn resulted in improved performance on sensory tasks by creating a more accurate representation of stimuli.

Cortical states are usually defined based on the correlated activity of neuronal populations[36]. Several studies have reported enhanced sensory responses in desynchronised states due to lower noise correlations[52,53]. During anaesthesia, sleep or quiet restful states, the cortex is in a deactivated state characterised by the presence of synchronised activity. In contrast, desynchronous firing is more prevalent during alert, attentive, and active behavioural conditions. At the cellular level, Muscarinic receptor inhibition blocks the slow membrane potential fluctuations associated with whisking[8] and

these modulations of membrane potential phases can in turn shape the stimulus-dependent and stimulus-independent correlations[12]. The stimulus-independent correlations between vS1 neurons, known as noise correlations, are typically higher in quiescent wakefulness compared to active exploration and whisking (Supplementary Movies 1, 2)[19,32]. In this study, we observed increased correlations between pairs of vS1 neurons when M1 was blocked. This is consistent with an increased synchronised activity with cholinergic inhibition[12]. On the other hand, M1 activation induced desynchrony among neurons, indicating enhanced capacity for information coding at the population level. Previous studies have reported enhanced sensory evoked responses during desynchronised states due to reduced noise correlations[52,53]. In line with this, our data showed that M1 activation enhanced whisker-evoked responses (Figs. 1, 2) and reduced synchrony at the population level (Fig. 3). This is consistent with studies highlighting that desynchronised states enhance response reliability in the somatosensory[54], visual[12] and auditory[38] cortices. Inhibitory interneurons play an important role in modulating this desynchronisation[55]. During quiet wakeful states, fast-spiking parvalbumin (PV) interneurons show synchronised firing[56] but are desynchronised during awake and attentive states. Somatostatin-expressing (SST) interneurons are also critical for precise synchronised firing in vS1[57]. The excitability of PV and SST interneurons can be differentially regulated by muscarinic activation, with muscarinic activation exhibiting atypical hyperpolarising or biphasic responses in interneurons[58,59]. Therefore, it is likely that M1 alters synchrony by differentially modulating the inhibitory drive in the vS1 neurons.

Cholinergic input to the cortex can vary dynamically depending on the level of arousal. For example, higher ACh levels are present in the cortex during awake attentive states as compared to lower ACh levels during quiet wakeful or anaesthetised states[60,61]). The M1 potentiator, BQCA, used in this study increases the affinity of endogenous ACh to M1 receptors by binding to an allosteric site[62]. Under anaesthesia, the enhancement of the evoked response by M1 was less pronounced compared to the enhancement observed during awake states (Supplementary Fig. 5). This suggests that M1-mediated modulations are more potent during active awake states, possibly due to the increased release of ACh from the BF. However, the presence of M1-mediated modulations during reduced cholinergic tone (i.e. during anaesthetised states) suggests a fundamental role for M1 receptors in sensory processing. This is consistent with a recent study where cholinergic signalling through muscarinic activation facilitated auditory evoked activity in response to passive auditory stimuli, outside of any attentional context[63]. Together, this indicates that a basal level of M1 activation plays an important role in passive sensory processing across sensory modalities. Therefore, it is likely that M1 modulates sensory processing through a unified mechanism that is preserved across sensory systems. It is interesting to note that the effects of M1 activation observed in this study - improved task performance, increased evoked response, improved neuronal response reliability and desynchronised firing – closely resemble the changes observed during enhanced attention[51].

The perceptual response to sensory input changes dynamically based on behavioural demands. Previous studies in rodents have shown that modulations in cortical state, such as those induced through muscarinic receptors, produce changes in behavioural performance[34,64,65]. Based on this literature and as confirmed in our single cell recording and $Ca^{2+}$ imaging data, we predicted that modulations in M1 receptor activity would directly influence behavioural responses. We used the whisker vibration detection paradigm as an ideal model to study sensory processing due to the ecological relevance of the whisker pathway in rodent

behaviour. Using their whiskers, rodents can be trained to learn complex behavioural tasks, such as discriminating textures[18,66,67], discriminating vibrations[68,69], and localising objects[70,71]. In this study, we found that activating M1 produces an enhancement in the detection of vibrissal stimuli which was accompanied by a reduction in the response times (Fig. 4d). We found informative parallels between the neuronal response functions and the behavioural response functions mediated by M1 modulations. One key observation in the neuronal data was an enhanced evoked response in the absence of a significant change in baseline firing (Fig. 1e and Fig. 2f). Our mice exhibited a similar pattern in the form of their enhanced detection rates in the absence of changes to their false alarm behaviour (Supplementary Fig. 7b). Overall, the M1-induced enhancement of the neuronal response functions at the population level (Fig. 1d and Fig. 2e) showed remarkable parallels to the M1-induced enhancement of the behavioural response functions (Fig. 4c). When an animal is actively engaged in the task, there is more cholinergic input into vS1 from the BF[8] and applying BQCA to vS1 potentiates the cholinergic response. In the visual system, muscarinic activation enhanced how the visual cortex responds to stimuli presented within an attended visual receptive field[3]; and enhanced visual discrimination performance by engaging M1[72]. Muscarinic activation also facilitated auditory evoked responses in the auditory cortex[63]. Our results are also consistent with a recent study that implemented an operant discrimination learning paradigm, where M1 inhibition reduced acquisition and consolidation[73]. Together these findings suggest that M1 critically modulates behavioural performance across various modalities.

Despite the systematic findings on M1-induced neuronal gain modulation and enhanced behavioural responses, we observed some level of heterogeneity among neurons. Some responsive neurons showed reduced evoked response after M1 activation and some quiescent neurons (<10%) exhibited an enhancement in evoked response after M1 inhibition (Fig. 2d, #Neuron 55–60). A cell-type specific modulation by M1 could explain this heterogeneity[74]. M1 is predominantly expressed on cortical pyramidal neurons (Supplementary Fig. 1a) with a subset of inhibitory GABAergic PV interneurons (Supplementary Fig. 1b) and SST interneurons[57] also expressing M1. Parallels with the nicotinic system suggest that a small subset of neurons expressing a receptor can be very functionally relevant in feedback, feedforward or disinhibitory microcircuits[74]. An important circuit motif for state modulation in the vS1 is through disinhibition[75,76] consisting of PV, SST, and vasoactive intestinal peptide (VIP) interneurons. When VIP interneurons receive cholinergic projections from the BF[77], they remove the inhibition on layer 2/3 pyramidal neurons exerted by SST interneurons; thereby inducing a more active desynchronised state, similar to the M1-mediated desynchrony observed in this study. Therefore, it is likely that an M1-modulated disinhibitory microcircuit in layer 2/3 is responsible for this sensory sharpening (enhanced evoked response of excitatory pyramidal neurons). Future experiments will be necessary to determine how the cortical microcircuit is influenced by specific M1-mediated modulations of VIP, SST and PV interneurons during sensory processing.

## Methods

**Mice.** All experiments were performed on male and female C57Bl/6 J mice (4–12 weeks old) housed in air-filtered and climate-controlled cages on a 12–12 h dark/light reverse-cycle. All methods were performed in accordance with the protocol approved by the Animal Experimentation and Ethics Committee of the Australian National University (AEEC 2019/20 and 2022/16). Mice had access to food and water *ad libitum* except in behavioural experiments where mice were water restricted. The weight and overall health of all animals was monitored on a regular basis.

**Juxtacellular electrophysiology.** Mice were anaesthetised with a urethane/chlorprothixene anaesthesia (0.8 g/kg and 5 mg/kg, respectively) and placed on a heating blanket at 37 °C. They were head-fixed on a custom-made apparatus. The scalp was opened via a 5 mm midline incision. After removing the scalp fascial tissue, a metal head plate was screwed to the posterior part of the skull and fixed in position with super glue and cemented subsequently. Once the cement had set, a 2 mm craniotomy was made above the right primary somatosensory cortex. The coordinates of the barrel cortex were marked as 1.8 mm posterior and 3.5 mm lateral to Bregma. The vasculature of the animal was also used as a reference to shortlist appropriate regions for recording.

Borosilicate glass pipettes were made by using a micropipette puller (P-97, Sutter Instruments) and custom-made programs. The recording pipettes had a tip diameter of ~0.5–1 μm (impedance of 6–10 MΩ) and the infusion pipettes had a diameter of ~20–30 μm with longer taper tips. The recording pipette was attached with glue to the infusion pipette on a custom-made stereotaxic setup with a tip-to-tip distance of 30–50 μm (Fig. 1a, Kheradpezhouh et al.[78]).

The recording pipette was filled with a 2% neurobiotin (in Ringer's solution). The infusion pipette was attached to a syringe pump (CMA402, Harvard Apparatus, Holliston, MA, USA) and filled with either artificial cerebrospinal fluid (aCSF), M1 receptor agonist Benzyl quinolone carboxylic acid (BQCA, 10 μM) or antagonist Telenzepine dihydrochloride (TD, 1 μM). The infusion pipette applied either aCSF, BQCA or TD at a flow rate of 2.5 μl/min. The pipette pair was positioned above the craniotomy and lowered using a micromanipulator. When the pipette pair reached the dura, 1 nA ON/OFF pulses (200 ms, 2.5 Hz) in current-clamp mode were applied. As the pipette touched the dura, Z-position of the micromanipulator was noted down for identifying the neuronal depth. The pressure in the recording pipette was maintained at 300 mmHg at this stage to avoid blockage of the pipette. After passing the dura, the pressure inside the recording pipette was reduced to 10–15 mm Hg, and the pipette was advanced at a speed of ~2 μm/s while searching for neurons. The resistance was continuously monitored using the current clamp mode of a Dagan Amplifier (BVC-700A). Proximity to a neuron was observed by fluctuations in recording voltage and an increase in the resistance of the pipette ( > 5-fold increase). At this step, the pressure was reduced to 0 mm Hg and juxtacellular (loose-cell attached) recording was performed. A custom-made MATLAB code provided the stimulus and recorded the neuronal response. Multiple recording session were made for all three conditions, aCSF, BQCA and TD. A total of 23 neurons were recorded from 17 mice.

At the end of the recording session, the recording pipette was moved closer to the neuron, which is indicated by an increase in the amplitude of voltage being measured (>2 mV). To further identify the morphology of a subset of neurons with neurobiotin by applying current pulses increasing in steps from 1 to 8 nA at 200 ms duration. Successful loading was observed by broadening the AP spikes and a high frequency, tetanic like neuronal firing[21,79].

*Vibrissal stimulation.* A custom-made MATLAB code generated a pseudorandom sequence of stimulus amplitudes and acquired electrophysiological data through a data acquisition card (National Instruments, Austin, TX) at a sampling rate of 64 kHz. A wire mesh (2 cm × 2.5 cm) attached to a piezoelectric

stimulator (Morgan Matroc, Bedford, OH) was slanted parallel to the animal's left whisker pad (~2 mm from the surface of the snout) on the contralateral side, making sure that the whiskers reliably engage with the mesh. A consistent distance was maintained between the mesh and the face of the mouse. The whisker stimuli were composed of single Gaussian deflection amplitudes of 0, 25, 50, 100 and 200 μm for juxtacellular recordings, and 0, 25, 50, 100 and 250 μm for $Ca^{2+}$ imaging. For behavioural experiments, the vibration stimulus was a train of discrete Gaussian deflections at amplitudes of 0, 15, 30, 60, or 120 μm. Each deflection lasted for 15 ms and was followed by a 10 ms pause before the next deflection.

**GCamp7f transfection and surgeries**. Mice were briefly anesthetized with isoflurane (~2% by volume in O2) and placed on a heating pad blanket (37 °C, Physitemp Instruments). Isoflurane was passively applied through a nose mask at a flow rate of 0.4–0.6 L/min. The level of anaesthesia was monitored by the respiratory rate, and hind paw and corneal reflexes. The eyes were covered with a thin layer of Viscotears liquid gel (Alcon, UK). During this surgical procedure, the scalp of anaesthetised mice was opened along the midline using scissors and a 3-mm craniotomy was performed over vS1 while keeping the dura intact. Expression of the $Ca^{2+}$ indicator GCaMP7f (Addgene, AAV1.-Syn.GCaMP7f.WPRE.SV40) was achieved by stereotaxic injection of AAV virus. GCaMP7f was injected in the cortex at a depth of 230–250 μm from the dura at 4–6 sites (with four 32-nL injections per site separated by 2–5 min at the rate of 92 nLs$^{-1}$). Following injections, a cranial window was covered using a 3 mm glass coverslip (0.1 mm thickness, Warner Instruments, CT). The animals were also implanted with a titanium headbar posterior to the cranial window, and a cannula (26 Gauge, Protech International Inc.) for microinjections of aCSF, BQCA or TD, immediately lateral to the cranial window. A small well was created around the cranial window using dental cement to allow water immersion for 2-Photon imaging. A thin layer of a silicon sealant (Kwik-Cast, World Precision Instruments, USA) was applied to cover all parts of the cranial window and skull.

**2-Photon $Ca^{2+}$ imaging**. 3–4 weeks following the injection of GCaMP7f, the animal was transferred to a two-photon imaging microscope system (ThorLabs, MA) with a Cameleon (Coherent) TiLSapphire laser tuned at 920 nm. The laser was focused onto layer 2/3 cortex through a 16x water-immersion objective lens (0.8NA, Nikon), and $Ca^{2+}$ transients were obtained from neuronal populations at a resolution of 512 × 512 pixels (sampling rate, ~30 Hz) (×16, 0.58NA). Laser power was adjusted between 40–75 mW depending on GCaMP7f expression levels. All image acquisition was via ThorImage (ThorLabs, MA) and frames were synchronised with the stimulus presentation via the data acquisition card. For awake recordings, mice were gradually habituated to the head-fixation apparatus—an acrylic tube with a custom-made headpost to allow head-fixation. After 3–4 days of habituation, mice were head-fixed in the apparatus and imaged.

To study the effect of M1 modulation on neuronal response to whisker stimulation, aCSF, BQCA or TD were applied to this region of the cortex through the implanted cannula by switching between syringe pumps (CMA402, Harvard Apparatus, Holliston, MA, USA) at a speed of 2 μl/min. All videos were processed using the Python Suite2P package (https://github.com/cortex-lab/Suite2P) for motion correction and semi-automated ROI detection was performed in conjunction with ImageJ. The mean background neuropil was subtracted from each neuron's $Ca^{2+}$ trace using a custom MATLAB script. The change in fluorescence ($\Delta F/F_0$) was quantified by using $F_0$ as the mean fluorescence for each recording session.

**Training and behavioural task**. Mice implanted with the headbar and cannula were allowed to recover for 1 week, and placed on a water restriction schedule. The animals were gradually habituated to the experimenter and the head-fixation apparatus. The duration of placing the animal in the tube was increased gradually and once the animals were adequately habituated with the setup, they were held in position near the headpost with the help of homoeostatic forceps, gradually increasing the duration of the hold. At each session, the mice were also presented with a 5% sucrose reward. Mice received unrestricted water for 2 h immediately following the training sessions.

When the mice were well habituated to the setup, the first stage of training began where the animals received a reward for every lick. A vibration pulse (1 s) followed each lick. This allowed the mice to lick reliably and get the sucrose reward. In the next stage of training, the mice were presented with a vibration till they licked the reward spout three times to claim the sucrose reward, after which a 60 s no-go period was enforced. In the last stage of training, the stimulus was either 120 μm (go) or 0 μm (no-go) with a variable inter-trial interval of 5–10 s. After mice learnt this version (above ~85% correct), The vibration duration reduced from 1 s in the first stage to 400 ms. The mice gradually learnt to lick the reward spout in response to a vibration of any amplitude (0, 20, 40, 80 or 120 μm). Stimulus amplitudes were pseudo randomised in blocks of 5 trials, with each block having all stimulus amplitudes. This produced an average of 240 trials per recording session (5 stimulus amplitudes X 48 trial blocks). Each session was repeated 5 times for every drug condition (aCSF, BQCA and TD).

A custom-made capacitive 'lick-port', connected to an Arduino UNO board (Duinotech Classic, Cat#XC4410), was used to deliver a sucrose reward and register licks. The lick-port was consistently positioned within reach of the mouth, ~0.5 mm below the lower lip and ~5 mm posterior to the animals' snout. The capacitive voltage was sent to data acquisition card and a threshold determined the presence or absence of a lick.

**Immunohistochemistry**. At the end of the experiment, the animals were euthanised by an intraperitoneal injection of lethabarb (150 mg/kg). After opening the abdomen and chest medially, the heart was perfused with chilled normal saline followed by 4% paraformaldehyde in phosphate buffered saline (PBS) and the brain was harvested. The brain was fixed in 4% paraformaldehyde in PBS at 4 °C overnight. After sequential rehydration with 10–30% Sucrose, the brain was sliced using a cryostat and incubated with streptavidin Alexa Fluor488 conjugate (Thermo Fisher Scientific, Waltham, MA, USA) overnight on a shaker at 4 °C. For immunostaining of PV interneurons and pyramidal neurons (Supplementary Fig. 1), 100 μm thick coronal sections were permeabilised with PBS containing 1% Triton-X and 0.1% Tween 20 for 2–3 h. To block non-specific binding sites slices were incubated in a blocking solution (0.25% Triton-X, 2% Bovine Serum Albumin in PBS), for 20–30 min at room temperature. Slices were then incubated with primary antibodies for M1 (Goat anti-M1 AChR, Abcam, Cat#ab77098, dilution 1:200), PV (Rabbit anti-PV, Abcam, Cat# ab11427, dilution 1:250) and CaMKII (Rabbit anti-CaMKII, Abcam, Cat#ab32678, dilution 1:250) added to blocking solution overnight at 4 °C on a shaker. The following day, slices were washed and incubated with their respective secondary antibodies (Donkey Anti-goat 568, Abcam Cat#ab175704, dilution 1:1000 or Donkey Anti-goat 488, Abcam Cat#ab150129, dilution 1:1000 for M1; Donkey Anti-rabbit 568, Abcam

Cat#ab175475, dilution 1:2000 for PV and Goat anti-rabbit 488, Cat#ab150077, dilution 1:1500 for CaMKII) for 3–4 h. Slices were then stained with 4′, 6-diamidino-2-phenylindole (DAPI) to stain cell nuclei and mounted with Immu-Mount mountant (Thermo Scientific, Cat# 9990402) onto microscope slides.

**Drug application**. In all experiments, we employed specific pharmacological agents to modulate M1 receptors: M1 receptor potentiator, Benzyl Quinolone Carboxylic acid (BQCA, 10 μM); M1 receptor agonist, Cevimeline Hydrochloride (5 μM); and M1-specific inhibitors, Telenzepine Dihydrochloride (TD, 1 μM for anesthetized mice and 5 μM for awake mice); and Dicyclomine Hydrochloride (5 nM). The drugs were delivered via a syringe pump (CMA402, Harvard Apparatus, Holliston, MA, USA) at a consistent rate of 2 μl/min. The order of drug application was pseudorandomized to minimise potential order effects. Between the switches in drug administration, a 2-min waiting period was implemented before initiating recordings (see Fig. 1a, bottom panel). For the anaesthetised recordings, we reduced the TD concentration to 1 μM. This was to avoid the potential toxicity of TD due to reduced CSF clearance rate under anaesthesia.

**Anaesthetic protocol**. Mice were anaesthetised with an intra-peritoneal injection of urethane/chlorprothixene (0.8 g/kg and 5 mg/kg, respectively) and placed on a heating blanket at 37 °C. Anaesthesia was maintained at a surgical depth, as assessed by the lack of spontaneous movement, lack of muscle tone in the jaw, face, and body, the absence of a paw withdrawal reflex in response to a toe pinch, as well as a rapid, shallow breathing rate, without gasping[80]. At this stage, mice stop initiating whisker movements, eye and eyelid movements[81]. Anaesthetic depth was measured regularly throughout the recording session to ensure that all recordings took place in a plane of surgical anaesthesia. Lastly, mice were confirmed to be at a surgical depth of anaesthesia immediately following the recording, by the absence of a paw withdrawal reflex in response to a toe pinch.

**Neuronal data analysis**. The spikes in each trial were extracted by applying a threshold for each neuron on the bandpass-filtered signal acquired during each recording session, using a custom-written MATLAB code. Neuronal firing rates were calculated by counting the number of spikes in each trial over a 50 ms window after the whisker-stimulus onset (0 ms). For every neuron and every stimulus amplitude and condition (aCSF, BQCA or TD), the mean firing rate (spikes/s) of 30 trials was reported. The latency of neuronal response was calculated as the timing of the first evoked spike in a 100 ms time bin, where the average firing rate was significantly higher than the baseline. A paired t-test was used to validate if the recorded neurons had a significant whisker evoked response, by comparing the baseline response in a 100 ms pre-stimulus window to the evoked firing response in a 100 ms post-stimulus window. This comparison was made across all 3 conditions – aCSF (baseline versus evoked), BQCA (baseline versus evoked) and TD (baselines versus evoked) to include the responsive neurons in any condition.

The Fano factor was calculated by dividing the variance (standard deviation squared) by the mean of the firing rate. $Fano\ Factor = \frac{\sigma^2}{Mean}$.

The best-fitting line, slope and intercept for each neuron were calculated and plotted using GraphPad Prism version 8.1.2.

**Noise correlations**. To calculate the noise correlation coefficient between neuron pairs, we computed the cross correlogram (using the MATLAB 'xcorr' function, and 'coeff' normalisation) of neuron pairs during periods of spontaneous activity in the absence of stimulus presentation. This allowed us to capture any stimulus-independent correlations or noise correlations in neuronal activity. Cross-correlation measurements were normalised to vary between 0 and 1. For each cell pair, the mean fluorescence activity ($\Delta F/F_0$) was correlated. The maximum height of the correlogram at lag 0 was taken as a measure of correlation strength.

**Behavioural analysis**. The lick rate was calculated by subtracting the licks in a 400 ms pre-stimulus window from the licks in the post-stimulus reward window of 1 s. Hit trials were defined as the presence of at least one lick in the post-stimulus window and no licks 400 ms before stimulus onset and were used to calculate the detection rate. To account for changes in motivation and engagement throughout the task, we excluded blocks of trials where the mice licked at 0 μm stimulus (false alarm). Here, the stimulus present trials (20, 40, 80 or 120 μm) were used to calculate the detection rate in each block (0—No stimulus detected, 1—All 4 stimulus intensities detected correctly). d-prime was computed for all trials by norminv (Hit rate)—norminv (False alarm rate), where norminv is the inverse of the cumulative normal function[82]. Hits and False alarm rates were truncated between 0.01 and 0.99.

**Statistics and reproducibility**. Relevant statistical analyses, p-values, and n-numbers are reported in figure legends and results section. Group data were presented as mean ± standard error of the mean (SEM). Statistical significance was determined using MATLAB and GraphPad Prism version 8.1.2. A Kolmogorov–Smirnov test was used to assess the normality of data. Data that did not pass the normality test used subsequent non-parametric statistic tests. A Friedman test with Dunn's multiple comparisons was used for non-parametric data that required a pairwise comparison across different conditions (aCSF, BQCA and TD). A Wilcoxon signed-rank test was used for pairwise comparisons of non-parametric data.

The order of drug application was pseudorandomized to minimise potential order effects. Ca$^{2+}$ Imaging and Behavioural experiments consisted of 5 recording sessions for every drug condition (aCSF, BQCA and TD).

**Reporting summary**. Further information on research design is available in the Nature Portfolio Reporting Summary linked to this article.

## Data availability
The data generated in this study is available at: https://osf.io/rd8b4/[83].

## Code availability
The codes for analysis have been published in an open-access format (https://doi.org/10.5281/zenodo.10360563)[84]. This is available at: https://github.com/MishraWricha/Cholinergic-M1-receptors-in-sensory-procesing.

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

## Acknowledgements

The experiments were supported by an Australian Research Council (ARC) Discovery Project (DP1701009), an NHMRC ideas grant (GNT1181643) and the ARC Centre of Excellence for Integrative Brain Function (ARC Centre Grant, CE140100007).

## Author contributions

W.M., E.K. and E.A. conceived and designed the project. W.M. performed the experiments. W.M., E.K. and E.A. analysed the data. W.M., E.K. and E.A. wrote the manuscript. All of the authors edited the manuscript and approved the final version.

## Competing interests

The authors declare no competing interests.
