## [Peer Review File · Communications Biology]

Reviewers' comments:

Reviewer #1 (Remarks to the Author):

The manuscript by Mishra et al. entitled Activation of M1 cholinergic receptors in mouse somatosensory cortex enhances information processing and improves detection behavior is thorough and relevant for the field. Even though the role of muscarinic M1 receptors in neural activity is very well known to increase the excitability, how they precisely reshape neural dynamics during sensory events in primary cortex is not well understood. The present work is convincing and shed more light on the complex dialog between cholinergic and somatosensory systems.

The authors describe the enhancing of sensory processing and sensory detection mediated by intracortical infusion of highly selective muscarinic M1 receptor allosteric modulator BQCA. To assess the pharmacological effect of M1 on sensory processing, the authors use a plethora of techniques, combining difficult in vivo juxtacellular recordings with paired pipette, calcium imaging and behavioral detection task in awake mice. Spike and behavioral analyses are compelling, revealing not only how M1 modulates and reshapes sensory responses in barrel cortex by reducing jittering and increasing response magnitude but also the functional relevance of this modulation, decreasing reaction time and increasing detection rate in awake mice. However, to improve clarity and reproducibility the authors need to address some minor issues.

Introduction

1: In my opinion the Introduction is extremely short and does not introduce the reader to the aims of the manuscript or to the importance of the muscarinic M1 receptor for sensory processing, despite of the extensive literature regarding the function of muscarinic and nicotinic receptors and their modulation in different cortices. The abstract and the introduction are undistinguishable from each other, I recommend rewriting the introduction section.

There are important and recent relevant papers about cholinergic modulation of sensory processing missing in the reference list, like for example: Jimenez-Martin et al., 2021; Chavez-Coira et al., 2018 and 2019.

Results

Line 64." ... M1 is prominently expressed in layer 2/3 and layer 5". Are there differences in the colocalization of M1/PV or M1/CAMKII between Layer 2/3 and Layer 5? If the authors have this data, it would be very helpful for the understanding of M1 role in sensory processing to know if there are differences between layer 2/3 and layer 5.

1 Fig1a. It would be very helpful to add a schematic depicting the stimulation protocol.

2 It would be more informative if authors could provide raster examples of more vibration amplitudes.

3 Which statistical test was used to validate if the recorded neurons were significantly responsive?

4 Among all recorded neurons did the authors find neurons decreasing in firing rate during BQCA or increasing during TD conditions? If so, it would be more informative to add pie plots counting the proportion of increased and decreased neurons in firing rates at each specific amplitude and show in which layer.

5 Fig1f. What does the vertical and horizontal black line indicate? I assume they indicate the mean and the standard deviation as in G. please add this information in the caption for panel f, or indicate it applies for all panels.

6 In figure 1 F G and H dots do not seem to be normally distributed as well as in figure 3d and in some cases median analysis would better represent the population tendency. Did the authors perform a normality test before statistical analysis? If so, indicate which test?

7 Were all neurons recorded in the same layer? If so, this should be clearly indicated in the results and discussed accordingly, if not, it would be very informative to provide an analysis of the firing rate changes in different layers under different conditions.

8 Fig2F, Under BQCA conditions the overall effect is an increase of calcium signaling, however, and interestingly some neurons are activated under TD condition. Is there any mechanism that would

explain this opposing effect? Did the authors observe something similar during the anesthetized juxtacellular experiments?

9 While the concentration of BQCA was the same for anesthetized and awake experiments, the concentration of TD was different, for anesthetized mice was 1 micromolar, however for awake experiments the concentration was 5 micromolar. Please indicate the reason of different concentrations.

10 Figure 4: I find confusing the behavioral paradigm, was the vibration of the piezo, the cue of the upcoming event? In Fig 4a there is an orange vertical line, is this the cue? I recommend the authors to redesign the protocol figure to make it clearer.

11 Figure 4b under vibration 0 micrometers the licking rate is slightly decreased, is this correct? It seems that under BQCA conditions this drop is stronger, what would be very interesting. Do the authors know if the decrease in liking rate was significantly different under BCQA? If I correctly understood the paradigm, this would not only mean an improve in detection rate but also discrimination.

12 I recommend adding a histology image showing the correct position of the cannulas in barrel cortex.

Material and methods

13 If I correctly understood a metal grid was attached to the piezo electric to stimulate multiple whiskers and produced a vibration that could also deflect the whiskers in different amplitudes from 0 – 250 micrometers. Was this vibration itself the cue during the behavioral task?

14 It would be helpful to provide a supplementary figure showing the awake detection task setup.

15 Supplementary figure 1b merged, I am concerned with this figure, it seems to me that color brightness of the red channel is not correctly adjusted since almost all PV positive neurons shown in red channel are not visible in merged. How did the authors analyzed colocalization?

Discussion

16 Line 386 -388 "Under anaesthesia, the M1-induced enhancement in the evoked response and baseline activity was less prominent as compared to the awake condition (Supplementary Fig. 3)." This statement is confusing and needs to be rephrased. Figure 3 shows an increase in baseline activity while in awake there is a slight decrease.

17 The discussion is very well written and informative however it would be helpful to discuss this results in the context of phasic acetylcholine release and volume transmission hypothesis, see: <https://www.ncbi.nlm.nih.gov/pmc/articles/PMC2699581/>

Reviewer #2 (Remarks to the Author):

The manuscript by Wricha and colleagues presents an investigation into the activation of M1 receptors in the vS1 region, which enhances information processing capabilities and improves animal performance in a vibrotactile detection task. This study emphasizes the crucial role of the cholinergic system within the somatosensory cortex for efficient information processing. The authors employed electrophysiology, 2-photon imaging, and behavioral experiments to demonstrate the significant role of M1 receptors in sensory processing. Moreover, they discovered that the cholinergic system enhances information transmission by desynchronizing a subpopulation of cortical neurons through the activation of M1 receptors. These findings offer valuable insights into the underlying mechanisms through which the cholinergic system modulates cortical neuronal activity during information processing. Based on these findings, I recommend this manuscript for publication in *Communications Biology*. However, there are some concerns that should be addressed.

Major concerns

1. Regarding the necessity of M1 receptors, it would be valuable to investigate whether the excitability of cortical neurons in response to whisker stimulation is altered after cKO of M1 receptors. Alternatively, it is recommended to employ other M1 receptor agonists to validate the findings and avoid potential off-target effects of BQCA.
2. In Figure 3a, the authors should include information regarding the spatial locations of the 6 selected neurons. Additionally, considering the utilization of the *hsyn-GCaMP7f*, the ROIs may include inhibitory neurons that exhibit synchronization. It would be helpful to clarify the criteria used for selecting these neurons.
3. In Figure 4, the data primarily assess the detection performance of mice following modulate of M1. However, there is a lack of information regarding the activity of cortical neurons after stimulation. It would be beneficial to investigate whether the improvement in detection performance aligns with the findings and hypotheses presented in Figure 2. Specifically, in go/no go paradigm with whisker stimulation, it is important to explore whether the activation of M1 receptors leads to an increase in the number of activated neurons and their responses, thereby enhancing the detection performance of the mice.

Minor concerns

1. The images in Figure S1, representing L2/3, are currently not displaying the desired level of detail. To address this, it would be beneficial to provide larger-scale brain slice staining that covers cortical layers L1-6, followed by zooming in to specifically highlight the staining in L2/3.
2. On page 5, in Figure 1D, it is stated that neuron N=23. However, it would be helpful to specify the number of mice from which these neurons were obtained. This comment also applies to other figures.
3. On page 10, in Figure 2C, the trace represents the average of how many regions of interest (ROIs), and from how many mice? Please provide details regarding the number of ROIs used to generate the trace, as well as the number of mice involved.
4. It would be helpful to clarify the methods of drug administration. Specifically, provide information on how long in advance the drugs were injected and the dosage used.
5. In Figure 4f, the bar graph depicts the average results of the trials. It would be helpful if you could provide information on the total number of trials conducted per mouse and consider incorporating scatter plot markers to visually represent individual data points.

Reviewer #3 (Remarks to the Author):

Review of Mishra et al: Activation of M1 cholinergic receptors in mouse somatosensory cortex

The paper by Mishra et al is an impressive study of the effects of a subset of cholinergic receptor activation in sensory cortex. The techniques used range from extracellular (juxtacellular) physiology to measure single neuronal responses, to in vivo multiphoton imaging, to measure correlated activity among neurons, to behavior. The results from each of the measurements converge nicely to present a complete story of the effects of muscarinic activation of sensory cortex, and work very nicely toward a theory of cholinergic arousal effects. The manuscript is well organized and clearly written.

The experiments are well designed, using proper statistical analysis and sufficient statistics. However, I feel that there are a number of issues that must be addressed, which would could significantly improve the manuscript. (See specific comments below).

1. I think that the manuscript needs more detail about the anesthesiology. The level of urethane + anesthesia can vary over time, and that variability can affect the level of anesthesia and sensory responses—and especially, the degree of correlation between responses. How was the level of anesthesia monitored? (If I missed this point in the manuscript, I apologize.) If (hopefully) with ECoG, then I would like to see the traces of shown together with both electrophysiology and imaging data. This would add considerable information to the study, as one might crosscorrelate a general cortex ECoG with the images in the barrel field for a local/global correlation. (If not, the authors should address the issue in the methods section).

2. Under urethane+ anesthesia, animals can move their whiskers, depending on anesthesia depth. This affects the stimuli themselves: if the stimulus hits the whisker differently, even at a slightly different angle each time, it will affect the variability of the response. How did the authors monitor this?

3. The authors need to show primary electrophysiological data: the juxtacellular traces corresponding to Figure 1. This should be included in Figure 1, not in the supplementary information. Examples of traces from a single neuron responding to different stimulus amplitudes would be a good idea. Further discussion about this point below.

4. The crosscorrelations shown in figure 4B are a major point of the study. However, crosscorrelations are an average of the synchronous activity over time. If possible (i.e. if the data collection lasted over a significantly long time to allow for sufficient statistics, a time-dependent cross-correlation example and summary (similar to Figures 3B, D, &F) would greatly enhance the manuscript. The procedure for the Joint PSTH can be found in Aertsen et al, J. Neurophysiol (1989) 61:5 900-917. I also believe that Matlab software exists on the web. This method provides information about the changes in correlation over time, and would greatly enhance our understanding of the temporal properties of the muscarinic neuromodulation.

5. The major point of the study is the effects of M1 modulation of neuronal and behavioral activity. The authors do an admirable job of showing increased response in identified pyramidal neurons, desynchronization, and sensory enhancement. However, they physiological mechanism of this is not really discussed in depth.

As the authors are no doubt aware, the membrane potential of cortical neurons under urethane + anesthesia (as well as awake animals) fluctuates between 2 states. The difference in the states results from different levels of correlated spatiotemporal synaptic background activity. The response

properties of those neurons vary between the states. It is quite possible that the significant changes recorded under M1 agonists results from a shift of the membrane potential state transition patten: the neuron then spends more time in the "Up State". In these circumstances, neuronal activity is less synchronized, spike latencies are lower, etc. This could be an explanation of the effect of the M1 effect that was demonstrated in the study. In the absence of intracellular recording date, it is not possible to precisely know the state of the neuron. However, some indications can be found. For example, if the authors did record the ECoG, that provides indications of the neuronal states: the high-frequency portion of the ECoG is correlated with the Up State. It would be a simple matter to do a frequency analysis of the ECoG in the various experimental conditions to see if this is the case, or if the changes due to M1 activation are due to other neuronal properties. The JPST histogram analysis suggested in comment #4 would also reveal the exact changes in the temporal pattern of correlated firing activity over time, with different changes expected if the effects are due to modulation of state activity, as opposed other possible mechanisms. In any event, this issue should be briefly addressed in the discussion.

Reviewers' comments:

Reviewer #1 (Remarks to the Author):

The manuscript by Mishra et al. entitled Activation of M1 cholinergic receptors in mouse somatosensory cortex enhances information processing and improves detection behavior is thorough and relevant for the field. Even though the role of muscarinic M1 receptors in neural activity is very well known to increase the excitability, how they precisely reshape neural dynamics during sensory events in primary cortex is not well understood. The present work is convincing and shed more light on the complex dialog between cholinergic and somatosensory systems.

The authors describe the enhancing of sensory processing and sensory detection mediated by intracortical infusion of highly selective muscarinic M1 receptor allosteric modulator BCQA. To assess the pharmacological effect of M1 on sensory processing, the authors use a plethora of techniques, combining difficult in vivo juxtacellular recordings with paired pipette, calcium imaging and behavioral detection task in awake mice. Spike and behavioral analyses are compelling, revealing not only how M1 modulates and reshapes sensory responses in barrel cortex by reducing jittering and increasing response magnitude but also the functional relevance of this modulation, decreasing reaction time and increasing detection rate in awake mice. However, to improve clarity and reproducibility the authors need to address some minor issues.

Introduction

1: In my opinion the Introduction is extremely short and does not introduce the reader to the aims of the manuscript or to the importance of the muscarinic M1 receptor for sensory processing, despite of the extensive literature regarding the function of muscarinic and nicotinic receptors and their modulation in different cortices. The abstract and the introduction are undistinguishable from each other, I recommend rewriting the introduction section.

There are important and recent relevant papers about cholinergic modulation of sensory processing missing in the reference list, like for example: Jimenez-Martin et al., 2021; Chavez-Coira et al., 2018 and 2019.

Thank you for this point. We have now revised the introduction thoroughly and have added more background including the papers suggested by the reviewer. The new sections are in lines 41 - 59:

The cholinergic system is well suited to coordinate neural activity across different modalities, as it provides a widespread and diffuse innervation of the cortex (Zaborszky et al., 2015). Acting through various subtypes of ACh receptors (AChRs), ACh controls neuronal excitability and firing rate, by hyperpolarising or depolarising target neurons (Gulledge et al., 2007; Gulledge & Stuart, 2005). Cholinergic stimulation also reduces noise correlations and membrane potential fluctuations (Eggermann et al., 2014; Meir et al., 2018). These functions are predominantly mediated by postsynaptic muscarinic AChRs (mAChRs) that

activate the G α q-signalling cascade and consequently increasing neuronal excitability (Shinoe et al., 2005).

Consistent with these functions, activation of mAChRs is known to enhance the sensory representations across modalities of vision, audition and somatosensation (Jose L. Herrero et al., 2017; Jimenez-Martin et al., 2021; Minces et al., 2017; S. Soma et al., 2013). As a well-studied example, ACh increased the response of neurons in the visual cortex to stimulus contrast (S. Soma et al., 2013) and enhanced orientation tuning of cortical neurons (Zinke et al., 2006). In the somatosensory cortex, mAChR modulation depolarised neuronal membrane potentials, and enhanced stimulus-evoked responses to deflections applied to the whiskers (Jimenez-Martin et al., 2021).

Despite growing evidence on the muscarinic neuromodulation, it is not clear how mAChRs shape the encoding of sensory inputs in single neurons and neuronal ensembles, ultimately determining the perceptual responses to those inputs.

Results

Line 64." ... M1 is prominently expressed in layer 2/3 and layer 5". Are there differences in the colocalization of M1/PV or M1/CAMKII between Layer 2/3 and Layer 5? If the authors have this data, it would be very helpful for the understanding of M1 role in sensory processing to know if there are differences between layer 2/3 and layer 5.

Thank you for this suggestion. We analysed the colocalisation of M1/PV and M1/CAMKII across layers as suggested. Layer 2/3 and layer 5 exhibited the highest degree of colocalisation. However, the difference between these two layers was not statistically significant. We have now included this analysis in the Supplementary Figs. 1 and 2 (below) and refer to it in the main Results.

Supplementary Fig. 2: M1 expression on the vS1 inhibitory neurons. a) Co-localisation of M1 (green) with PV (red, a marker for a subtype of inhibitory interneurons) across layers of the vS1 (Scale bar = 100µm). b) Inset - Co-localisation of M1 and PV in Layer 5 and Layer 2/3 of the vS1 (Scale bar = 60µm). c) Pearson's correlation coefficients depicting the co-localisation of M1 and PV in Layer 5 and Layer 2/3.

Supplementary Fig. 1: M1 expression on vS1 excitatory neurons. a) Co-localisation of M1 (red) with CaMKII (green, a marker for excitatory neurons) across layers of the vS1 (Scale bar = 100µm). b) Inset - Co-localisation of M1 and CaMKII in Layer 5 and Layer 2/3 of the vS1 cortex (Scale bar = 50µm). c) Pearson's correlation coefficients depicting the co-localisation of M1 and CaMKII in Layer 5 and Layer 2/3.

1 Fig1a. It would be very helpful to add a schematic depicting the stimulation protocol.

We have now added a schematic to new Fig1a (below) to demonstrate the sequence of manipulations and their duration in the protocol. The relevant text is also added to the Methods section (lines 860 - 864):

The drugs were delivered via a syringe pump (CMA402, Harvard Apparatus, Holliston, MA, USA) at a consistent rate of 2 µl/min. The order of drug application was pseudorandomized to minimize potential order effects. Between the switches in drug administration, a 2-minute waiting period was implemented before initiating recordings (see Fig 1a, bottom panel).

2 It would be more informative if authors could provide raster examples of more vibration amplitudes.

We have added raw data in the form of raster plots in new Fig. 1c (Updated Fig. 1 shown above).

3 Which statistical test was used to validate if the recorded neurons were significantly responsive?

We apologise for not indicating the statistical test earlier. This is now added to the text in the Methodology section (lines 882 - 886):

'A paired t-test was used to validate if the recorded neurons had a significant whisker evoked response, by comparing the baseline response in a 100ms pre-stimulus window to the evoked firing response in a 100 ms post-stimulus window. This comparison was made across all 3 conditions – aCSF (baseline versus evoked), BQCA (baseline versus evoked) and TD (baselines versus evoked) to include the responsive neurons in any condition.'

4 Among all recorded neurons did the authors find neurons decreasing in firing rate during BQCA or increasing during TD conditions? If so, it would be more informative to add pie plots counting the proportion of increased and decreased neurons in firing rates at each specific amplitude and show in which layer.

During the Calcium Imaging experiments, we did observe a small subpopulation of neurons that showed a reduction in evoked response with BQCA. We have now discussed this more explicitly in the Discussion section reporting the proportions (lines 486 - 490):

'Despite the systematic findings on M1-induced neuronal gain modulation and enhanced behavioural responses, we observed some level of heterogeneity among neurons. Some responsive neurons showed reduced evoked response after M1 activation and some quiescent neurons (<10%) exhibited an enhancement in evoked response after M1 inhibition (Fig. 2d, #Neuron 55-60).'

5 Fig1f. What does the vertical and horizontal black line indicate? I assume they indicate the mean and the standard deviation as in G. please add this information in the caption for panel f, or indicate it applies for all panels.

The reviewer is correct. We have now added this in the caption for Fig1 (lines 121 - 122) :

'For panels f), g) and h), the horizontal and vertical lines represent the mean and SEM respectively.'

6 In figure 1 F G and H dots do not seem to be normally distributed as well as in figure 3d and in some cases median analysis would better represent the population tendency. Did the authors perform a normality test before statistical analysis? If so, indicate which test?

We thank the reviewer for this comment. We have now added a test for normality and modified the stats where necessary. This has now been included in the Methods section under the Statistics subsection (lines 910 - 915) :

'A Kolmogorov-Smirnov test was used to assess the normality of data. Data that did not pass the normality test used subsequent non-parametric statistic tests. A Kruskal-Wallis test with Dunn's multiple comparison with Dunn's multiple comparisons was used for non-parametric data which required a comparison across different conditions (aCSF, BQCA and TD). A Wilcoxon signed-rank test for used for pairwise comparisons of non-parametric data.'

7 Were all neurons recorded in the same layer? If so, this should be clearly indicated in the results and discussed accordingly, if not, it would be very informative to provide an analysis of the firing rate changes in different layers under different conditions.

In new Fig. 1d we have now plotted the average neuronal response functions separately for Layer 2/3 and Layer 5 neurons.

8 Fig2F, Under BQCA conditions the overall effect is an increase of calcium signaling, however, and interestingly some neurons are activated under TD condition. Is there any mechanism that would explain this opposing effect? Did the authors observe something similar during the anesthetized juxtacellular experiments?

We thank the reviewer for this observation. The increase in evoked response after M1 inhibition is seen in a subset of neurons (<10%). We hypothesise that this atypical modulation of evoked response by M1 receptors is likely to be exhibited by Parvalbumin (PV)-expressing interneurons in the S1. Using PV-Cre mice, we are currently investigating this hypothesis in our lab. We have included the reviewer's observation and the hypothesised circuit motif in the discussion (lines 486-504):

'Despite the systematic findings on M1-induced neuronal gain modulation and enhanced behavioural responses, we observed some level of heterogeneity among neurons. Some responsive neurons showed reduced evoked response after M1 activation and some quiescent neurons (<10%) exhibited an enhancement in evoked response after M1 inhibition (Fig. 2d, #Neuron 55-60). A cell-type specific modulation by M1 could explain this heterogeneity (Gasselin et al., 2021). M1 is predominantly expressed on cortical pyramidal neurons (Supplementary Fig. 1a) with a subset of inhibitory GABAergic PV interneurons (Supplementary Fig. 1b) and SST interneurons (Chen et al., 2015) also expressing M1. Parallels with the nicotinic system suggest that a small subset of neurons expressing a receptor can be very functionally relevant in feedback, feedforward or disinhibitory microcircuits (Gasselin et al., 2021). An important circuit motif for state modulation in the cortex is through disinhibition (Gainey et al., 2018; S. Lee et al., 2013) consisting of PV, SST, and vasoactive intestinal peptide (VIP) interneurons. When VIP interneurons receive cholinergic projections from the BF (Zagha & McCormick, 2014), they remove the inhibition on layer 2/3 pyramidal neurons exerted by SST interneurons; thereby inducing a more active desynchronised state, similar to the M1-mediated desynchrony observed in this study. Therefore, it is likely that an M1-modulated disinhibitory microcircuit in layer 2/3 is responsible for this sensory sharpening (enhanced evoked response of excitatory pyramidal neurons). Future experiments will be necessary to determine how the cortical microcircuit is influenced by specific M1-mediated modulations of VIP, SST and PV interneurons during sensory processing.'

9 While the concentration of BQCA was the same for anesthetized and awake experiments, the concentration of TD was different, for anesthetized mice was 1

micromolar, however for awake experiments the concentration was 5 micromolar. Please indicate the reason of different concentrations.

We used the concentration consistent with earlier experiments reported in the literature. The concentration of TD was reduced to 1 μ M in anaesthetised experiments to avoid toxicity due to a reduced CSF clearance rate. Cerebrospinal fluid (CSF) circulation and clearance is active during wakefulness but significantly impaired during states of anaesthesia (Gakuba et al. 2018). We have made our logic explicit in the Methods section (lines 863-865):

For the anaesthetised recordings, we reduced the TD concentration to 1 μ M. This was to avoid potential toxicity of TD due to reduced CSF clearance rate under anaesthesia.

10 Figure 4: I find confusing the behavioral paradigm, was the vibration of the piezo, the cue of the upcoming event? In Fig 4a there is an orange vertical line, is this the cue? I recommend the authors to redesign the protocol figure to make it clearer.

We apologise for the confusion in this figure. The orange line indicated the onset of the vibration stimulus and not the cue. We have now modified the figure (new Fig. 4a, shown below) such that the orange rectangle indicates the vibration stimulus (in its entirety of 400 ms and not just the onset) and added a grey rectangle to indicate the reward window of 1s.

11 Figure 4b under vibration 0 micrometers the licking rate is slightly decreased, is this correct? It seems that under BQCA conditions this drop is stronger, what would be very interesting. Do the authors know if the decrease in liking rate was significantly different under BCQA? If I correctly understood the paradigm, this would not only mean an improve in detection rate but also discrimination.

This is an interesting observation. However, the lick rate on 0 μm trials was not significantly different between the control and BQCA condition ($p = 0.1501$, Mann-Whitney U test) or in the TD condition ($p = 0.9871$, Mann-Whitney U test). We believe that the 0-stimulus lick rates under BQCA appear lower due to the higher pre-stim lick rates under that condition.

12 I recommend adding a histology image showing the correct position of the cannulas in barrel cortex.

As recommended, we have now added the histology figure in Supplementary Fig. 5a (Shown below). Thank you.

Supplementary Fig. 5: **a)** A histology image showing the position of the implanted cannula (Dil, red) along with GCaMP7f expression (Green) in the vS1. **b)** Characterisation of M1 modulation on neuronal population using 2-photon Calcium imaging in anaesthetised animals. Changes in $\Delta F/F_0$ measured in a 1 s window after stimulus onset in the aCSF (black), BQCA (green) and TD (magenta) conditions for all stimulus amplitudes. The dots represent the mean $\Delta F/F_0$; the error bars represent the SEM (n=410, 4 mice). **c)** M1 modulation of the baseline (left) and maximum evoked (right) response. Every dot represents a neuron. Black lines and error bars indicate the mean and SEM across neurons (n = 410, 4 mice, *p<0.01, Kruskal-Wallis Test with Dunn's multiple comparison).

Material and methods

13 If I correctly understood a metal grid was attached to the piezo electric to stimulate multiple whiskers and produced a vibration that could also deflect the whiskers in different amplitudes from 0 – 250 micrometers. Was this vibration itself the cue during the behavioral task?

Yes, the reviewer is correct. There was no separate cue in the behavioural task and the piezo lasted for 400ms and marked the beginning of the 1s reward window. We have now modified the schematic in Fig. 4a for better clarity.

14 It would be helpful to provide a supplementary figure showing the awake detection task setup.

We have now included a picture of the set up taken under infrared lighting during the task (Supplementary Fig. 7a).

a**b**
Supplementary Fig. 7: **a)** The awake detection task set-up: An example head-fixed mouse engaged in the detection task. A cannula is implanted over the vS1 for drug perfusion during the task. A wire mesh connected to a piezoelectric stimulator is used to stimulate the whiskers at different amplitudes and a reward spout delivers a drop of sucrose in hit trials. **b)** The average false alarm rates across 6 mice show no significant changes with M1 modulations ($n = 6$, Kruskal-Wallis Test with Dunn's multiple comparison).

15 Supplementary figure 1b merged, I am concerned with this figure, it seems to me that color brightness of the red channel is not correctly adjusted since almost all PV positive neurons shown in red channel are not visible in merged. How did the authors analyzed colocalization?

We have now adjusted the red channel and this has improved the quality of the histological images in Supplementary Figs. 1 and 2 (shown below). We thank the reviewer for noting this.

Supplementary Fig. 1: M1 expression on vS1 excitatory neurons. **a)** Co-localisation of M1 (red) with CaMKII (green, a marker for excitatory neurons) across layers of the vS1 (Scale bar = 100µm). **b)** Inset - Co-localisation of M1 and CaMKII in Layer 5 and Layer 2/3 of the vS1 cortex (Scale bar = 50µm). **c)** Pearson's correlation coefficients depicting the co-localisation of M1 and CaMKII in Layer 5 and Layer 2/3.

Supplementary Fig. 2: M1 expression on the vS1 inhibitory neurons. a) Co-localisation of M1 (green) with PV (red, a marker for a subtype of inhibitory interneurons) across layers of the vS1 (Scale bar = 100µm). b) Inset - Co-localisation of M1 and PV in Layer 5 and Layer 2/3 of the vS1 (Scale bar = 60µm). c) Pearson's correlation coefficients depicting the co-localisation of M1 and PV in Layer 5 and Layer 2/3.

Discussion

16 Line 386 -388 "Under anaesthesia, the M1-induced enhancement in the evoked response and baseline activity was less prominent as compared to the awake condition (Supplementary Fig. 3)." This statement is confusing and needs to be rephrased. Figure 3 shows an increase in baseline activity while in awake there is a slight decrease.

We agree, the reference to baseline changes was confusing. We have now corrected the statement (lines 443-447):

'Under anaesthesia, the enhancement of the evoked response by M1 was less pronounced compared to the enhancement observed during awake states (Supplementary Fig. 5). This suggests that M1-mediated modulations are more potent during active awake states, possibly due to the increased release of ACh from the BF.'

17 The discussion is very well written and informative however it would be helpful to discuss this results in the context of phasic acetylcholine release and volume transmission hypothesis, see: <https://www.ncbi.nlm.nih.gov/pmc/articles/PMC2699581/>

This was a good suggestion. We have now added a discussion point about phasic ACh release (lines 384-396) -

'Here, we showed that M1 activation significantly enhanced the sensory-evoked responses and reduced the trial-to-trial variability of these responses. This was illustrated by an increase in the signal strength, a decrease in the Fano Factors of the firing rates and a decrease in the first-spike latencies. Our method of activating M1 through a local potentiator can be seen as a phasic release of ACh mediated by Basal Forebrain (BF) neurons, at the scale of seconds. Our findings are thus consistent with phasic ACh release and a

volume transmission hypothesis as the circuit mechanism underlying cognitive operations such as attention (Sarter et al., 2009).'

Reviewer #2 (Remarks to the Author):

The manuscript by Wricha and colleagues presents an investigation into the activation of M1 receptors in the vS1 region, which enhances information processing capabilities and improves animal performance in a vibrotactile detection task. This study emphasizes the crucial role of the cholinergic system within the somatosensory cortex for efficient information processing. The authors employed electrophysiology, 2-photon imaging, and behavioral experiments to demonstrate the significant role of M1 receptors in sensory processing. Moreover, they discovered that the cholinergic system enhances information transmission by desynchronizing a subpopulation of cortical neurons through the activation of M1 receptors. These findings offer valuable insights into the underlying mechanisms through which the cholinergic system modulates cortical neuronal activity during information processing. Based on these findings, I recommend this manuscript for publication in *Communications Biology*. However, there are some concerns that should be addressed.

Major concerns

1. Regarding the necessity of M1 receptors, it would be valuable to investigate whether the excitability of cortical neurons in response to whisker stimulation is altered after cKO of M1 receptors. Alternatively, it is recommended to employ other M1 receptor agonists to validate the findings and avoid potential off-target effects of BQCA.

We are thankful for this suggestion. We have now further validated the findings by performing experiments using a different M1 receptor agonist (Cevimeline Hydrochloride) and antagonist (Dicyclomine Hydrochloride), both selective to M1. We used juxtacellular electrophysiology for this experiment following the same methods already reported in the manuscript. As before, we characterised neuronal modulations by plotting average neuronal response function (Supplementary Fig. 3b), first-spike latencies (Supplementary Fig. 3c) as well as changes in baseline, maximum evoked response, and response range (Supplementary Fig. 3d). These results replicated earlier modulations. We now refer to this validation experiment in the Results section (lines 138-144).

To further validate the findings, we performed neuronal recordings using a different M1 receptor agonist (Cevimeline Hydrochloride, 5 μ M) and antagonist (Dicyclomine Hydrochloride, 5 μ M). The agonist and antagonist produced a qualitatively similar modulation of neuronal activity as observed earlier with BQCA and TD. As before, activation of M1 increased the baseline firing, the maximum evoked response and the response range (Supplementary Fig. 3).

2. In Figure 3a, the authors should include information regarding the spatial locations of the 6 selected neurons. Additionally, considering the utilization of the hsyn-GCaMP7f,

the ROIs may include inhibitory neurons that exhibit synchronization. It would be helpful to clarify the criteria used for selecting these neurons.

Thank you for this suggestion. We have now specified the spatial locations of the 6 selected neurons in Fig. 3a by highlighting the ROIs in Fig 2b (shown below).

The reviewer is correct in that the imaging data includes both excitatory and inhibitory neurons. Given that the whisker responsive inhibitory neurons form less than 10% of the imaged population, the data mostly captures the profile of the excitatory neurons. The only inclusion criteria that we used was a paired t-test to include responsive neurons

(both excited and inhibited by the stimulus). We have now better specified the criteria in the Methods section. We have also elaborated in the Discussion on the potential interaction of excitatory and inhibitory neurons within the circuit.

Methods (lines 882-886):

A paired t-test was used to validate if the recorded neurons had a significant whisker evoked response, by comparing the baseline response in a 100ms pre-stimulus window to the evoked firing response in a 100 ms post-stimulus window. This comparison was made across all 3 conditions – aCSF (baseline versus evoked), BQCA (baseline versus evoked) and TD (baselines versus evoked) to include the responsive neurons in any condition.

Discussion (lines 486-504):

Despite the systematic findings on M1-induced neuronal gain modulation and enhanced behavioural responses, we observed some level of heterogeneity among neurons. Some responsive neurons showed reduced evoked response after M1 activation and some quiescent neurons (<10%) exhibited an enhancement in evoked response after M1 inhibition (Fig. 2d, #Neuron 55-60). A cell-type specific modulation by M1 could explain this heterogeneity (Gasselín et al., 2021). M1 is predominantly expressed on cortical pyramidal neurons (Supplementary Fig. 1a) with a subset of inhibitory GABAergic PV interneurons (Supplementary Fig. 1b) and SST interneurons (Chen et al., 2015) also expressing M1. Parallels with the nicotinic system suggest that a small subset of neurons expressing a receptor can be very functionally relevant in feedback, feedforward or disinhibitory microcircuits (Gasselín et al., 2021). An important circuit motif for state modulation in the cortex is through disinhibition (Gainey et al., 2018; S. Lee et al., 2013) consisting of PV, SST, and vasoactive intestinal peptide (VIP) interneurons. When VIP interneurons receive cholinergic projections from the BF (Zagha & McCormick, 2014), they remove the inhibition on layer 2/3 pyramidal neurons exerted by SST interneurons; thereby inducing a more active desynchronised state, similar to the M1-mediated desynchrony observed in this study. Therefore, it is likely that an M1-modulated disinhibitory microcircuit in layer 2/3 is responsible for this sensory sharpening (enhanced evoked response of excitatory pyramidal neurons). Future experiments will be necessary to determine how the cortical microcircuit is influenced by specific M1-mediated modulations of VIP, SST and PV interneurons during sensory processing.'

3. In Figure 4, the data primarily assess the detection performance of mice following modulate of M1. However, there is a lack of information regarding the activity of cortical neurons after stimulation. It would be beneficial to investigate whether the improvement in detection performance aligns with the findings and hypotheses presented in Figure 2. Specifically, in go/no go paradigm with whisker stimulation, it is important to explore whether the activation of M1 receptors leads to an increase in the number of activated

neurons and their responses, thereby enhancing the detection performance of the mice.

This is an important point, and we have now made the parallel between the neuronal and the behavioural data more explicit. To allow the parallels to be made, the behavioural experiment was performed using exactly the same methods as the imaging and electrophysiological experiments. The same method of local vS1 drug perfusion was implemented. We followed the reviewer's suggestion and made parallels between our neuronal and behavioural data. One key observation in the neuronal data was an enhanced evoked response in the absence of a significant change in baseline firing. This was reflected both in electrophysiology data (Fig. 1e) as well as in calcium imaging data (Fig. 2f). After quantifying the behavioural data, we found a similar pattern in the false alarm behaviour across the 6 mice (Supplementary Fig. 7b). The M1-induced enhancement of the neuronal response functions at the population level (Fig. 1d and Fig. 2e) showed remarkable parallels to the M1-induced enhancement of the behavioural response functions (Fig. 4c). We thank the reviewer for this suggestion, and have now described these parallels between neuronal and behavioural data in the Results and Discussion sections.

In Results (lines 290-306):

Both electrophysiology and calcium imaging experiments demonstrated an M1-induced gain modulation of vS1 neurons. The enhanced sensory representations were evident at the level of single neurons (Fig. 1c, e, and Fig. 2d) and at the population level (Fig. 1d, Fig. 2e). The M1-induced gain modulation also resulted in an increased number of vS1 neurons that responded to the whisker input (25% under control to 31% under M1 activation). We therefore hypothesised that the enhanced representations in the vS1 would improve the mouse's ability to detect whisker stimuli. To determine the effect of M1 modulation on detection behaviour, we tested a simple detection task in awake head fixed mice while modulating M1 activity. As nocturnal animals, mice regularly use their whiskers to navigate and explore their surroundings. Depending on the behavioural state of the animal (active engagement or quiet wakefulness), the efficiency of sensory information processing is altered by neuromodulatory inputs like ACh (Eggermann et al., 2014). Here, we investigated whether the observed enhancement in sensory processing through M1 receptors is reflected in the behavioural performance of mice.

Using a similar modulation method to that used in the electrophysiology and imaging experiments, we implanted a cannula in the right vS1 of 6 mice and attached a headbar to allow head fixation...

In Discussion (lines 468-476):

We found informative parallels between the neuronal response functions and the behavioural response functions mediated by M1 modulations. One key observation in the neuronal data was an enhanced evoked response in the absence of a significant change in baseline firing (Fig. 1e and Fig. 2f). Our mice

exhibited a similar pattern in the form of their enhanced detection rates in the absence of changes to their false alarm behaviour (Supplementary Fig. 7b). Overall, the M1-induced enhancement of the neuronal response functions at the population level (Fig. 1d and Fig. 2e) showed remarkable parallels to the M1-induced enhancement of the behavioural response functions (Fig. 4c).

Minor concerns

1. The images in Figure S1, representing L2/3, are currently not displaying the desired level of detail. To address this, it would be beneficial to provide larger-scale brain slice staining that covers cortical layers L1-6, followed by zooming in to specifically highlight the staining in L2/3.

We apologise for the oversight. We have now improved the quality of the histological images and have added a larger-scale brain slice in new Supplementary Figs. 1 and 2 (shown below).

Supplementary Fig. 2: M1 expression on the vS1 inhibitory neurons. a) Co-localisation of M1 (green) with PV (red, a marker for a subtype of inhibitory interneurons) across layers of the vS1 (Scale bar = 100µm). b) Inset - Co-localisation of M1 and PV in Layer 5 and Layer 2/3 of the vS1 (Scale bar = 60µm). c) Pearson's correlation coefficients depicting the co-localisation of M1 and PV in Layer 5 and Layer 2/3.

Supplementary Fig. 1: M1 expression on vS1 excitatory neurons. a) Co-localisation of M1 (red) with CaMKII (green, a marker for excitatory neurons) across layers of the vS1 (Scale bar = 100µm). b) Inset - Co-localisation of M1 and CaMKII in Layer 5 and Layer 2/3 of the vS1 cortex (Scale bar = 50µm). c) Pearson's correlation coefficients depicting the co-localisation of M1 and CaMKII in Layer 5 and Layer 2/3.

2. On page 5, in Figure 1D, it is stated that neuron N=23. However, it would be helpful to specify the number of mice from which these neurons were obtained. This comment also applies to other figures.

As suggested, we have now added the number of mice in all figures where applicable.

3. On page 10, in Figure 2C, the trace represents the average of how many regions of interest (ROIs), and from how many mice? Please provide details regarding the number of ROIs used to generate the trace, as well as the number of mice involved.

The trace in Fig. 2c represents the average of 1 ROI for 30 trials. This ROI is now highlighted by the white arrow in Fig. 2b. We have also included this information in the legend for Fig 2c (lines 228-231):

c) Changes in fluorescence ($\Delta F/F_0$) in response to the 250 μm whisker deflection, under the control (aCSF), M1 activation (BQCA) and M1 inhibition (TD) conditions for the ROI shown in b (white arrow, n=30 trials).

4. It would be helpful to clarify the methods of drug administration. Specifically, provide information on how long in advance the drugs were injected and the dosage used.

All drugs were applied locally, intracortical through a syringe pump connected to a cannula implanted over the vS1. The information about the concentration of drugs used and the timing of the injections are now added in the Methods section under subsection Drug application (lines 857-863):

In all experiments, we employed specific pharmacological agents to modulate M1 receptors: M1 receptor potentiator, Benzyl Quinolone Carboxylic acid (BQCA, 10 μM); M1 receptor agonist, Cevimeline Hydrochloride (5 μM); and M1-specific inhibitors, Telenzepine Dihydrochloride (TD, 1 μM for anesthetized mice and 5 μM for awake mice); and Dicyclomine Hydrochloride (5 nM). The drugs were delivered via a syringe pump (CMA402, Harvard Apparatus, Holliston, MA, USA) at a consistent rate of 2 $\mu\text{l}/\text{min}$. The order of drug application was pseudorandomized to minimize potential order effects. Between the switches in drug administration, a 2-minute waiting period was implemented before initiating recordings (see Fig 1a, bottom panel).

5. In Figure 4f, the bar graph depicts the average results of the trials. It would be helpful if you could provide information on the total number of trials conducted per mouse and consider incorporating scatter plot markers to visually represent individual data points.

We thank the reviewer for this suggestion. Scatter plot markers have now been added in new Fig. 4f (shown below). The individual data points added to new Fig. 4f indicate the average detection rate of each animal for 5 recording sessions. Each session comprised of 240 trials (5 stimulus amplitudes X 48 trial blocks). We have now added this information in the Methods section under subsection Training and behavioural task as follows (lines 831-833):

'Stimulus amplitudes were pseudo randomised in blocks of 5 trials, with each block having all stimulus amplitudes. This produced an average of 240 trials per recording session (5 stimulus amplitudes X 48 trial blocks). Each session was repeated 5 times for each drug condition (aCSF, BQCA and TD).'

Reviewer #3 (Remarks to the Author):

Review of Mishra et al: Activation of M1 cholinergic receptors in mouse somatosensory cortex

The paper by Mishra et al is an impressive study of the effects of a subset of cholinergic receptor activation in sensory cortex. The techniques used range from extracellular (juxtacellular) physiology to measure single neuronal responses, to in vivo multiphoton imaging, to measure correlated activity among neurons, to behavior. The results from each of the measurements converge nicely to present a complete story of the effects of muscarinic activation of sensory cortex, and work very nicely toward a theory of cholinergic arousal effects. The manuscript is well organized and clearly written.

The experiments are well designed, using proper statistical analysis and sufficient statistics. However, I feel that there are a number of issues that must be addressed, which would could significantly improve the manuscript. (See specific comments below).

1. I think that the manuscript needs more detail about the anaesthesiology. The level of urethane + anesthesia can vary over time, and that variability can affect the level of anesthesia and sensory responses—and especially, the degree of correlation between responses. How was the level of anesthesia monitored? (If I missed this point in the manuscript, I apologize.) If (hopefully) with ECoG, then I would like to see the traces of shown together with both electrophysiology and imaging data. This would add considerable information to the study, as one might crosscorrelate a general cortex ECoG with the images in the barrel field for a local/global correlation. (If not, the authors should address the issue in the methods section.

We regularly monitored the level of anaesthesia through breathing rate, corneal and paw withdrawal reflexes and by monitoring the absence of any whisking action. We applied a 10% top up to urethane in presence of any reflexes or in case of whisking. In practice, urethane elicited a long-lasting stable surgical plane of anaesthesia (6+ hours). We did not implement a separate ECoG recording, but did analyse the degree of correlation over time in our recording electrodes and found it to be stable. We have included a quantification of correlations in Supplementary Video 1 to show these correlations over time. We have also added specific details of anaesthesiology in the Methods section under Anaesthetic protocol (lines 867-874).

Mice were anaesthetised with an intraperitoneal injection of urethane/chlorprothixene (0.8 g/kg and 5 mg/kg, respectively) and placed on a heating blanket at 37°C. Anaesthesia was maintained at a surgical depth, as assessed by the lack of spontaneous movement, lack of muscle tone in the jaw, face, and body, the absence of a paw withdrawal reflex in response to a toe pinch, as well as a rapid, shallow breathing rate, without gasping (Soma, 1983). At this stage, mice stop initiating whisker movements, eye and eyelid movements (Bharioke et al, 2022). Anaesthetic depth was measured regularly throughout the recording session to ensure that all recordings took place in a plane of surgical anaesthesia. Lastly, mice were confirmed to be at a surgical depth of anaesthesia immediately following the recording, by the absence of a paw withdrawal reflex in response to a toe pinch.

2. Under urethane+ anesthesia, animals can move their whiskers, depending on anesthesia depth. This affects the stimuli themselves: if the stimulus hits the whisker differently, even at a slightly different angle each time, it will affect the variability of the response. How did the authors monitor this?

In all our anaesthetised experiments, anaesthesia was maintained at a stable surgical depth, as assessed by the rate of breathing, the lack of spontaneous whisking, lack of

muscle tone in the jaw, face, and body, the absence of a paw withdrawal reflex in response to a toe pinch. We performed imaging under stable anaesthesia to enable a comparison between the M1-induced modulations of neuronal activity in the anaesthetised condition (Supplementary Fig. 5) to that of the awake head-fixed state (Fig. 2). The monitoring of anaesthesia and the lack of whisking are now made explicit in the Method section as indicated in our response to your previous comment. Following the reviewer's suggestion on cross-correlation analysis (please see comment 4 below) we made a similar video to demonstrate the stability of correlations over time during urethane anaesthesia (Supplementary Video 1).

3. The authors need to show primary electrophysiological data: the justacellular traces corresponding to Figure 1. This should be included in Figure 1, not in the supplementary information. Examples of traces from a single neuron responding to different stimulus amplitudes would be a good idea. Further discussion about this point below. This has now been added in Figure 1.

Thank you for this suggestion. We have added raw electrophysiological traces in Fig. 1b (shown below), which we believe has improved the figure significantly.

4. The crosscorrelations shown in figure 4B are a major point of the study. However, crosscorrelations are an average of the synchronous activity over time. If possible (i.e. if the data collection lasted over a significantly long time to allow for sufficient statistics, a time-dependent cross-correlation example and summary (similar to Figures 3B, D, &F) would greatly enhance the manuscript. The procedure for the Joint PSTH can be found in Aertsen et al, *J. Neurophysiol* (1989) 61:5 900-917. I also believe that Matlab software exists on the web. This method provides information about the changes in correlation over time, and would greatly enhance our understanding of the temporal properties of the muscarinic neuromodulation.

Refer to Cholinergic shaping of neural correlations, Minces et al.

This is a very good suggestion. To understand the changes in noise correlations over time, we have now added a time-dependent cross-correlation analysis across the entire recording session in the Supplementary Video 2. This provided an informative and clear visualisation of the M1-mediated changes in correlation over time.

We also implemented the Joint PSTH analysis as suggested to quantify the M1 induced modulations of signal correlation across pairs of neurons. This is now illustrated in the main text (lines 257-259) and in Supplementary Fig. 6 (below).

Supplementary Fig. 6: Joint peristimulus time-histogram (J-PSTH) analysis. **a)** For every trial, spikes from one neuron are arranged on the x- axis, and spikes from another neuron are arranged on the y-axis. Coincident spikes over the course of the trial are recorded in the J-PSTH matrix. This is repeated for every trial and presented as the heatmap. The 45-degree diagonal in this matrix is illustrated as a correlation time histogram on the right. **b)** The cumulative J-PSTH for all neuronal pairs in the control condition (aCSF, left), after M1 activation (BQCA, middle) and M1 inhibition (TD, right). The J-PSTH is normalised by the product of standard deviations of PSTHs (Aertsen et al., 1989). **c)** To correct for coincident spikes related to stimulus presentation, a shift predictor matrix (as in Aertsen et al., 1989) is subtracted from the raw J-PSTH, and the entire quantity is normalised by the product of standard deviations of PSTHs.

5. The major point of the study is the effects of M1 modulation of neuronal and behavioral activity. The authors do an admirable job of showing increased response in

identified pyramidal neurons, desynchronization, and sensory enhancement. However, they physiological mechanism of this is not really discussed in depth.

As the authors are no doubt aware, the membrane potential of cortical neurons under urethane + anesthesia (as well as awake animals) fluctuates between 2 states. The difference in the states results from different levels of correlated spatiotemporal synaptic background activity. The response properties of those neurons vary between the states. It is quite possible that the significant changes recorded under M1 agonists results from a shift of the membrane potential state transition patten: the neuron then spends more time in the "Up State". In these circumstances, neuronal activity is less synchronized, spike latencies are lower, etc. This could be an explanation of the effect of the M1 effect that was demonstrated in the study. In the absence of intracellular recording date, it is not possible to precisely know the state of the neuron. However, some indications can be found. For example, if the authors did record the ECoG, that provides indications of the neuronal states: the high-frequency portion of the ECoG is correlated with the Up State. It would be a simple matter to do a frequency analysis of the ECoG in the various experimental conditions to see if this is the case, or if the changes due to M1 activation are due to other neuronal properties. The JPST histogram analysis suggested in comment #4 would also reveal the exact changes in the temporal pattern of correlated firing activity over time, with different changes expected if the effects are due to modulation of state activity, as opposed other possible mechanisms. In any event, this issue should be briefly addressed in the discussion.

We thank the reviewer for their suggestion on the possible underlying mechanisms. Indeed conducting the analysis of J-PSTH and quantifications of correlation, as suggested in the previous comments, helps a better understanding of the M1-induced changes in the population dynamics. We have now discussed the potential underlying mechanisms at the cellular (lines 417-422) and circuit (lines 486-504) levels.

At the cellular level, muscarinic receptor inhibition has been shown to block the slow membrane potential fluctuations associated with whisking (Eggermann et al, 2014). These modulations of membrane potential phases are reflected in the synchronised activity at the population level. As the reviewer rightly pointed out, this suggests that M1 receptor activation is likely shifting the membrane potential of neurons in a way that consequently desynchronises population firing. Inhibitory interneurons may play an important role in modulating this desynchronisation (Middleton et al., 2012). During quiet wakeful states, fast-spiking parvalbumin (PV) interneurons show synchronised firing (Jang et al., 2020) but are desynchronised during awake and attentive states. Somatostatin-expressing (SST) interneurons are also critical for precise synchronised firing in the vS1 (Chen et al., 2015). The excitability of PV and SST interneurons can be differentially regulated by muscarinic activation, with muscarinic activation exhibiting atypical hyperpolarising or biphasic responses in interneurons (Cea-Del Rio et al., 2011; Pafundo et al., 2013). Therefore, it is likely that M1 alters synchrony by differentially modulating the inhibitory drive in the vS1 neurons.

REVIEWERS' COMMENTS:

Reviewer #1 (Remarks to the Author):

The manuscript by Mishra et al. has improved substantially in clarity, and the authors have addressed convincingly all my comments.

I recommend this manuscript for publication in Communications Biology.

Reviewer #2 (Remarks to the Author):

This manuscript has been well revised, and the authors have addressed almost all the concerns that were raised. I would like to recommend this manuscript for publication in Communications Biology.

Reviewer #3 (Remarks to the Author):

The authors have adequately addressed my concerns stated in the previous review.

Congratulations a a very nice study.

2. REVIEWERS' COMMENTS:

Reviewer #1 (Remarks to the Author):

The manuscript by Mishra et al. has improved substantially in clarity, and the authors have addressed convincingly all my comments.

I recommend this manuscript for publication in Communications Biology.

Reviewer #2 (Remarks to the Author):

This manuscript has been well revised, and the authors have addressed almost all the concerns that were raised. I would like to recommend this manuscript for publication in Communications Biology.

Reviewer #3 (Remarks to the Author):

The authors have adequately addressed my concerns stated in the previous review.

Congratulations a very nice study.

We would like to thank the three reviewers for their positive comments on our manuscript and for their detailed and constructive criticism.